DOI: 10.1038/s41467-018-05564-z | OPEN

# Methylation of all *BRCA1* copies predicts response to the PARP inhibitor rucaparib in ovarian carcinoma

Olga Kondrashova[1,2], Monique Topp et al.[#]

Accurately identifying patients with high-grade serous ovarian carcinoma (HGSOC) who respond to poly(ADP-ribose) polymerase inhibitor (PARPi) therapy is of great clinical importance. Here we show that quantitative *BRCA1* methylation analysis provides new insight into PARPi response in preclinical models and ovarian cancer patients. The response of 12 HGSOC patient-derived xenografts (PDX) to the PARPi rucaparib was assessed, with variable dose-dependent responses observed in chemo-naive *BRCA1/2*-mutated PDX, and no responses in PDX lacking DNA repair pathway defects. Among *BRCA1*-methylated PDX, silencing of all *BRCA1* copies predicts rucaparib response, whilst heterozygous methylation is associated with resistance. Analysis of 21 *BRCA1*-methylated platinum-sensitive recurrent HGSOC (ARIEL2 Part 1 trial) confirmed that homozygous or hemizygous *BRCA1* methylation predicts rucaparib clinical response, and that methylation loss can occur after exposure to chemotherapy. Accordingly, quantitative *BRCA1* methylation analysis in a pre-treatment biopsy could allow identification of patients most likely to benefit, and facilitate tailoring of PARPi therapy.

The development of therapy with poly(ADP-ribose) polymerase inhibitors (PARPi) has been a major advance in the treatment of high-grade serous ovarian carcinoma (HGSOC). PARPi are efficacious in HGSOCs with defective DNA repair by homologous recombination (HR) due to mutation in the breast and ovarian cancer predisposition genes *BRCA1* or *BRCA2* (*BRCA1/2*)[1]. When administered as maintenance therapy in the setting of platinum-sensitive relapsed HGSOC, PARPi prolong progression-free survival (PFS), with some patients deriving durable benefit for more than 3 years[2–6]. As a result, PARPi are now approved in both the treatment and maintenance settings in relapsed ovarian cancer (OC) by the European Medicines Agency (EMA) and the US Food and Drug Administration (FDA).

PARPi may also be relevant as targeted therapy for cancers with a range of defects in HR DNA repair beyond *BRCA1/2* mutation. In high-grade OC, HR defects caused by *BRCA1/2* mutations are present in 17–25% of cases, of which approximately ¾ are germline and ¼ are somatic[7–9]. Other HR pathway alterations have been documented in an additional 25% of HGSOC[7,9]. These HR defects include mutations in the HR pathway genes *RAD51C, RAD51D*, and *PALB2* (6–10%)[7,9,10], as well as methylation of *BRCA1* (7–17%)[7,11,12] or *RAD51C* (1.5–3%) promoters[11,13], which is generally mutually exclusive of *BRCA1/2* mutation[7,9,10].

Despite exciting clinical efficacy, one third of the patients with *BRCA1/2* mutant relapsed HGSOC fail to derive benefit from PARPi, with a higher failure rate observed with increasing platinum resistance. Even when patients do respond, the majority relapse within 12 months[3]. A well-defined PARPi resistance mechanism is restoration of HR function via secondary somatic mutations occurring within mutated *BRCA1/2* genes, resulting in re-institution of in-frame gene transcription[14,15]. Secondary mutations that revert primary *BRCA1/2* and *RAD51C/D* mutations have been described in HGSOC and prostate cancers in association with resistance to both platinum and PARPi therapy[10,16,17]. Improved understanding of HR defects beyond *BRCA1/2* mutations (both primary or secondary) is still required to allow more accurate targeting of PARPi therapy and design of strategies to abrogate PARPi resistance.

*BRCA1* promoter methylation was first noted 20 years ago in breast cancer[18], followed by reports in OC[19–22]. Methylation of CpG sites close to the *BRCA1* transcription start site[23,24] is associated with reduced *BRCA1* mRNA and protein[7,13,21,23]. Accordingly, one of the accepted mechanisms for functional BRCA1 loss involves methylation of one *BRCA1* allele combined with a loss of heterozygosity (LOH) event resulting in loss of the other *BRCA1* allele[21]. The impact of methylation of a single *BRCA1* copy, with retention or demethylation of another, on response to treatment remains unexplored. In support of *BRCA1* methylation conferring an HR defect, it has been associated with the same gene expression signature and copy number alterations observed in *BRCA1*-mutated HGSOC[25] and, more recently, with genomic signatures suggesting HR deficiency in breast cancer[26]. Contrary to these observations, unlike for *BRCA1/2* mutations, *BRCA1* methylation has not been shown to impact survival in patients with OC, with multiple studies failing to observe a significant improvement in overall survival upon stratification by *BRCA1* methylation status[7,11,27,28]. More recently in a clinical trial in triple-negative breast cancer, no benefit was observed for carboplatin in subjects with tumor-associated *BRCA1* methylation, compared with *BRCA1/2* mutation[29]. Further study of *BRCA1* methylation is required to reconcile these observations.

Use of PARPi therapy was previously proposed for cancers with *BRCA1* methylation[30]. A *BRCA1*-methylated breast cancer cell line displayed PARPi sensitivity; and *BRCA1* silencing as well

as PARPi sensitivity were abolished by the demethylating agent 5-azacytidine[31]. *BRCA1* methylation was also weakly associated with response to monotherapy with the PARPi rucaparib in the ARIEL2 Part 1 trial, but it was unclear which *BRCA1*-methylated cases would respond to treatment[32]. In contrast, in a study of long-term responders following maintenance therapy with PARPi after response to platinum, no long-term responders (>2 years) were found to have *BRCA1* methylation in their archival HGSOC[33]. Thus, the likelihood of PARPi response in patients with *BRCA1*-methylated HGSOC requires clarification.

Variable levels of *BRCA1* promoter methylation, ranging from 5 to 100%, have been previously reported in breast and OC samples, with most studies assigning "methylation" status to samples when as little as 5–15% methylation is detected[13,25,27,33]. In some cases, this is consistent with low neoplastic cellularity. However, the possibility that methylation of all *BRCA1* copies might be required to impact therapeutic outcome has not yet been addressed. Here we test the hypothesis that the zygosity status of *BRCA1* methylation (homozygous or hemizygous vs. heterozygous) may have an impact on PARPi or platinum response and may be affected by treatment pressure, allowing for the rapid development of drug resistance. The terms "homozygous" and "homozygosity" used to define the methylation status in this paper will cover all cases where unmethylated alleles are absent, regardless of the *BRCA1* copy number (Supplementary Fig. 1). Here we show that the rucaparib response of *BRCA1*-methylated OC cell lines and patient-derived xenografts (PDX) depends upon the *BRCA1* methylation zygosity. Further, we report quantitative methylation analysis of pre-treatment HGSOC samples from the ARIEL2 Part 1 PARPi trial, which is the only published clinical trial to date for which pre-treatment biopsies of cases documented to contain *BRCA1* methylation are available. In this clinical trial setting, we also demonstrate that *BRCA1* methylation zygosity correlates with rucaparib response.

## Results

**Genomic characterization of HGSOC PDX.** For this study, we have characterized PDX from 12 HGSOC patients, ten who were chemotherapy naive and two who had received multiple prior lines of therapy. Histologic assessment and WT1, PAX8, and p53 immunohistochemistry (IHC) staining confirmed that the PDX retained HGSOC features that were observed in the baseline carcinoma (Fig. 1a, Supplementary Fig. 2)[34]. The patient HGSOC and/or PDX whole tumor DNA samples were profiled using the Foundation Medicine T5a next-generation sequencing (NGS)-based test and RNA sequencing (RNA-seq). In addition, each PDX was also capture-sequenced for mutations in DNA repair pathway genes, in particular mutations that could cause HR deficiency, as previously described[9], and tested for *BRCA1* promoter methylation. Apart from the expected somatic mutations in *TP53*[35], mutations were also identified in *BRCA1/2* in four HGSOC, one of which was confirmed to be germline (#56; Fig. 1b, Supplementary Fig. 3a, Supplementary Data 1). Other events that are commonly detected in HGSOC included *RB1* mutation or deletion in six cases (#56, 19, 11, 169, 27, 80), *NF1* deletion in one case (#80), and *CCNE1* amplification in two cases (#29, 201). RNA-seq analysis, performed on the baseline patient HGSOC samples used to generate the PDX, confirmed the reduced expression of deleted genes and high expression of amplified genes (Fig. 1c, Supplementary Fig. 3b). In addition to deleterious *BRCA1/2* mutations in four HGSOC, *BRCA1* methylation was detected in four of the 12 HGSOC and in the corresponding PDX (#11, 62, 48, 169)[34]. The remaining four PDX were assigned an HR-DNA repair gene wild-type status, since no pathogenic mutations were detected in a curated set of HR pathway genes (Fig. 1b).

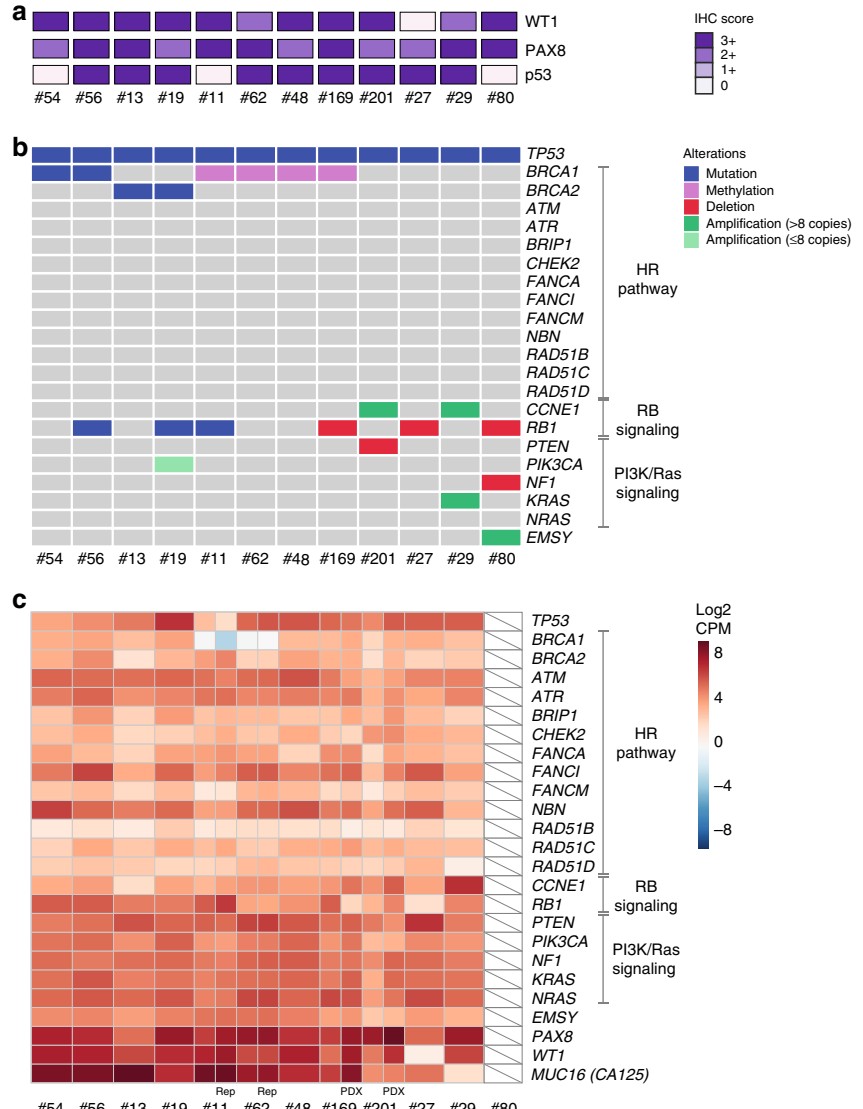

**Fig. 1** Genomic profiling of 12 HGSOC PDX. **a** IHC staining of PAX8, WT1, and p53 of passage one (T1) PDX tumors. Loss of p53 expression was observed for PDX #54 with a frameshift *TP53* mutation (p.G199fs*8), #11 with a nonsense *TP53* mutation (p.E198*), and #80 with a splice site *TP53* mutation (IVS6-1G>T). **b** Select genomic events detected by the Foundation Medicine T5a test, BROCA assay, and *BRCA1* promoter methylation testing. The Foundation Medicine T5a test was performed on PDX samples, except for case #48, where it was performed on patient HGSOC material. T5a test results and BROCA v4 assay results for PDX #11, #13, #27, #29, #56, and #62 were previously published[34]; BROCA v6 was performed for all other PDX[34]. **c** RNA-seq gene expression for genes with detected mutations or copy number changes. RNA-seq was performed on baseline patient HGSOC material samples. RNA-seq was also performed on PDX #169 and #201 samples, to verify expression levels observed in the matched HGSOC with suboptimal sample quality due to either low neoplastic cellularity or poor RNA quality (#80 inadequate quality); rep—RNA-seq library replicate

**Dose-dependent rucaparib responses in *BRCA1/2* mutant PDX**. To assess PARPi sensitivity, rucaparib was delivered by oral gavage 5 days a week for 3 weeks at one of the three dose levels—150, 300, or 450 mg kg$^{-1}$. As expected, three of four PDX that were HR deficient due to *BRCA1/2* mutations responded to rucaparib in vivo (Table 1, Supplementary Data 2). Some mice bearing PDX #19 or #56 obtained durable regressions lasting more than 80 days (Fig. 2a, b). Despite being tested in the chemo-naive/first-line setting, without prior exposure to chemotherapy or PARPi, variable dose-dependent responses were observed, and not all *BRCA1/2* mutant HGSOC PDX were equally sensitive to PARPi (Table 1, Fig. 2a, b, Supplementary Fig. 4a, b).

Two of four *BRCA1/2* mutant PDX responded to the lowest dose of rucaparib tested: *BRCA2* mutant PDX #19 (median time to harvest (TTH) 74 days vs. vehicle 22 days, $p = 0.012$, log-rank test,

$n = 4$, 21) and *BRCA1* mutant PDX #56 (median TTH 67 days vs. vehicle 15 days, $p = 0.003$, log-rank test, $n = 5$, 16) at 150 mg kg$^{-1}$. The chemo-naive *BRCA2* mutant PDX #13 had a statistically significant response to both 300 and 450 mg kg$^{-1}$ rucaparib, with median TTH of 81 days for rucaparib 300 mg kg$^{-1}$ vs. 43 days for vehicle ($p = 0.01$, log-rank test, $n = 9$, 22), although regressions were not observed. Strikingly, PDX #54, with a pathogenic missense *BRCA1* BRCT domain mutation (c.5095C>T, p.R1699W), was refractory to rucaparib in the first-line setting (median TTH for rucaparib 300 mg kg$^{-1}$ 36 days vs. vehicle 32 days, $p = 0.9$, log-rank test, $n = 9$, 4), possibly due to HSP90-mediated stabilization of the mutant BRCA1 protein, as has been observed with other BRCT domain-mutant BRCA1 proteins[36]. In further experiments, DNA sequencing failed to detect any secondary mutations in either *BRCA1* or *BRCA2* in these four PDX at recurrence (Supplementary Data 3).

**Table 1 Responses observed in 12 HGSOC PDX to cisplatin and rucaparib treatment in vivo**

| PDX # | Baseline tumor | Patient response to platinum agents/PARP inhibitors[a] | HR gene defect | TTH vehicle | Cisplatin response in PDX | | | | Rucaparib (300 mg kg$^{-1}$) response in PDX | | | | Explored mechanisms of resistance to rucaparib in vivo |
|---|---|---|---|---|---|---|---|---|---|---|---|---|---|
| | | | | | Response | Median TTH (days) | Average TTP (days) | p-value | Response | Median TTH (days) | Average TTP (days) | p-value | |
| #54 | Chemo-naive | Platinum sensitive[a] **PARPi unknown** | BRCA1: c.5095C>T | 32 | Resistant SD | 78 | 50 | 0.010 | Refractory PD | 36 | 8 | 0.900 | No secondary mutations; BRCA1 structural reversion predicted |
| #56 | Chemo-naive | Platinum sensitive[a] **PARPi 2nd line single agent CR** | BRCA1: c.894_895delTG | 15 | Sensitive CR | >120 | 113 | <0.001 | Response SD | 95 | 53 | <0.001 | No secondary mutations |
| #13 | Chemo-naive | Platinum resistant[a] **No PARPi** | BRCA2: c.5517_5518delA | 43 | Resistant PR | >120 | 99 | <0.001 | Minimal response SD | 81 | 32 | 0.010 | No secondary mutations |
| #19 | Chemo-naive | Platinum sensitive[a] **PARPi unknown** | BRCA2: c.2323_2323delT | 22 | Sensitive CR | >120 | >120 | <0.001 | Response CR | >120 | >120 | <0.001 | No secondary mutations |
| #11 | Chemo-naive | Platinum sensitive[a] **PARPi 4th line single agent PR for 11 months** | BRCA1 methylation homozygous | 46 | Sensitive CR | >120 | >120 | <0.001 | Not assessed | – | – | – | No loss of methylation |
| #62 | Chemo-naive | Platinum sensitive[a] **No PARPi** | BRCA1 methylation homozygous | 18 | Resistant/ Refractory SD | 60 | 46 | <0.001 | Response SD | 71 | 50 | <0.001 | No loss of methylation |
| #48 | Pre-treated | Platinum resistant **PARPi 3rd line single agent refractory** | BRCA1 methylation heterozygous | 36 | Resistant SD | >120 | 43 | <0.001 | Refractory PD | 67 | 8 | 0.095 | No further loss of methylation |
| #169 | Pre-treated | Platinum refractory **No PARPi** | BRCA1 methylation heterozygous | 29 | Refractory PD | 67 | 8 | 0.077 | Refractory[b] PD | 36 | 8 | 0.924 | No further loss of methylation |
| #201 | Chemo-naive | Platinum sensitive **No PARPi** | HR-DNA repair gene wild type | 25 | Resistant PR | 99 | 57 | <0.001 | Refractory PD | 46 | 8 | <0.001[c] | – |
| #27 | Chemo-naive | Platinum sensitive[a,d] **No PARPi** | HR-DNA repair gene wild type | 22 | Resistant PR | 109 | 57 | 0.001 | Refractory PD | 36 | 8 | 0.887 | – |
| #29 | Chemo-naive | Platinum refractory[a] **No PARPi** | HR-DNA repair gene wild type | 25 | Refractory PD | 32 | 8 | 0.128 | Refractory PD | 32 | 8 | 0.306 | – |
| #80 | Chemo-naive | Platinum sensitive **No PARPi** | HR-DNA repair gene wild type | 53 | Sensitive CR | >120 | >120 | 0.021 | Refractory PD | 64 | 15 | 0.021[c] | – |

PDX were derived from the chemo-naive baseline patient HGSOC samples apart from PDX #48, derived from a patient who had undergone three prior chemotherapeutic regimens, and PDX #169, generated from ascites fluid (the only PDX in this study not to be derived from solid tumor) from a young woman whose HGSOC progressed 1 month after completing first-line therapy and was refractory to second-line platinum treatment. Bold—patient PARPi response
TTH time to harvest, TTP time to progression, SD stable disease, CR complete response, PR partial response, PD progressive disease
[a]As previously reported[34]
[b]Rucaparib 450 mg kg$^{-1}$
[c]No tumor regressions or stabilization of disease was achieved despite significant p-value
[d]Clinical trial involving standard chemotherapy with placebo/novel agent, followed by maintenance therapy with placebo/novel agent

**Variable rucaparib responses in PDX with *BRCA1* methylation**. In keeping with the proposed requirement of an HR defect for PARPi response, the four PDX derived from HGSOC lacking mutation of HR genes and *BRCA1* promoter methylation showed no evidence of tumor regression or disease stabilization with rucaparib (Table 1, Fig. 2c, d, Supplementary Fig. 4c, d). In contrast, variable rucaparib responses were observed in models with *BRCA1* methylation. Two chemo-naive baseline patient HGSOC samples and the corresponding PDX (#11 and #62), in which no pathogenic HR gene mutations were detected, were found to harbor *BRCA1* methylation by methylation-specific PCR (MSP) as previously reported (Fig. 1b)[34]. Furthermore, two baseline patient HGSOC samples obtained from patients who had received prior treatment in the clinic and their corresponding PDX (#48 and #169) were also found to harbor *BRCA1* methylation by MSP. No other pathogenic events in HR pathway genes were identified in these four HGSOC or corresponding PDX (Fig. 1b).

The methylation status of these PDX samples was re-assessed by both methylation-specific high-resolution melting (MS-HRM) and methylation-sensitive droplet digital PCR (MS-ddPCR). Seven co-methylated CpG sites of the *BRCA1* promoter region were assessed; MS-HRM ($-37$, $-29$, $-21$, and $-19$)[37] and ddPCR ($+14$, $+16$, and $+19$). Two modes of *BRCA1* methylation were observed, with (i) homozygous methylation on all *BRCA1* copies present and (ii) heterozygous methylation where both

methylated and unmethylated copies were observed (Fig. 3a, b, Supplementary Figs. 1 and 5). The two chemo-naive PDX (#11 and #62) were consistently found to harbor ~100% *BRCA1* methylation and hence were assigned a homozygous status. The two PDX from HGSOC from patients treated with multiple lines of prior therapy (#48 and #169) were consistently found to harbor around 50% methylation and, therefore, were assigned a heterozygous status. The presence of two peaks in the MS-HRM analysis indicated that methylation was concordant across the four sites, as molecules with partial methylation would have intermediate melting temperatures and form more complex patterns (Supplementary Fig. 5). Analysis of the matched source patient HGSOC samples was also consistent with homozygous (#11 and #62) and heterozygous (#48 and #169) methylation of the *BRCA1* promoter, although due to variable neoplastic purity in patient HGSOC samples, it was more challenging to estimate the proportion of methylated copies (Supplementary Data 4). RNA-seq, *BRCA1* quantitative reverse transcription PCR (qRT-PCR) and western blotting (WB) analysis showed markedly reduced BRCA1 expression in the two source patient HGSOC samples and matched PDX models with homozygous methylation, but not in the two with heterozygous methylation, further supporting that methylation, mutation, or loss of all copies is required for *BRCA1* silencing (Supplementary Fig. 6a–c, Supplementary Table 1)[26]. We assessed HR pathway activity by ex vivo

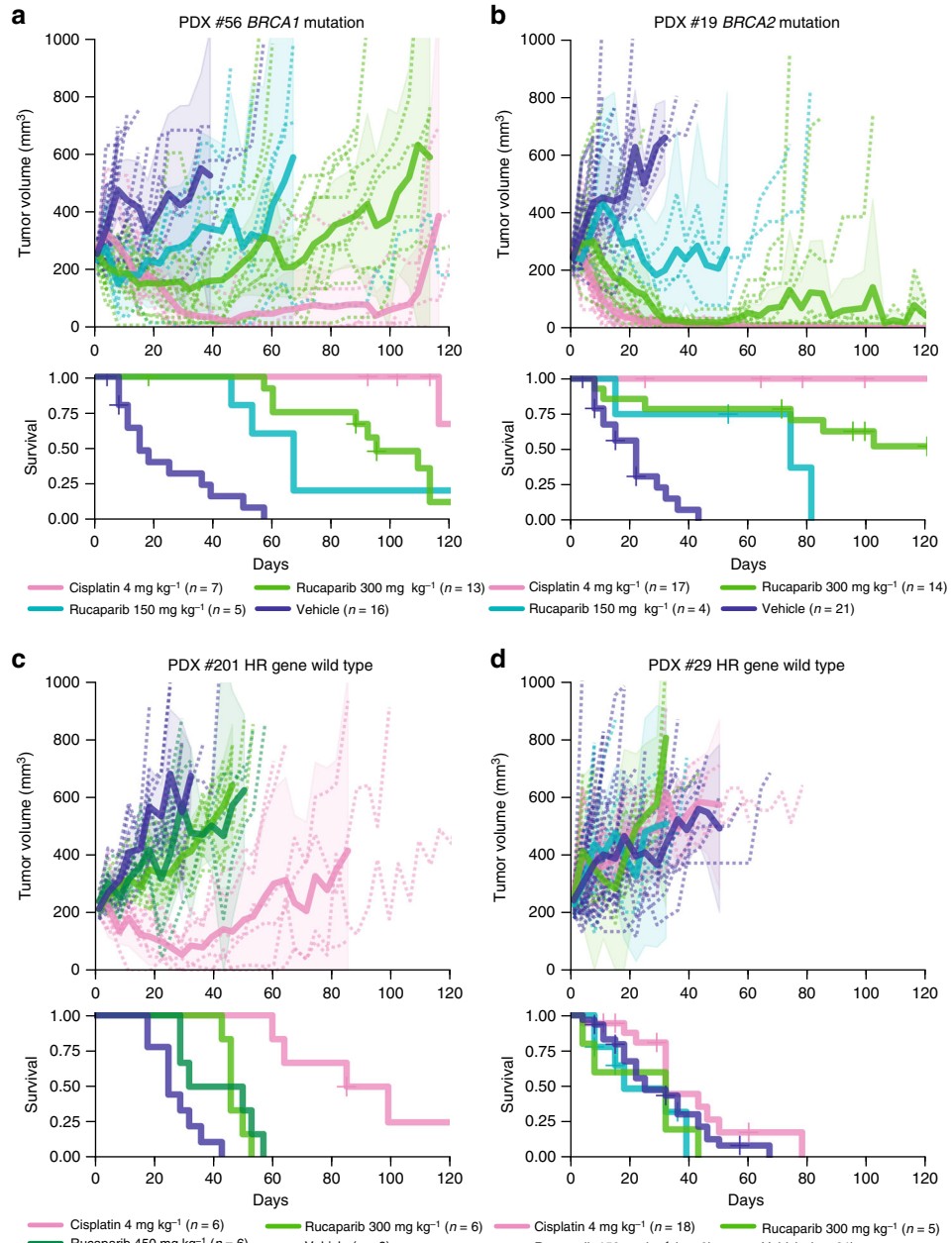

**Fig. 2** Cisplatin and rucaparib responses in *BRCA1/2* mutant and HR wild-type HGSOC PDX. Rucaparib and cisplatin response in **a** PDX #56 (*BRCA1* mutant); **b** PDX #19 (*BRCA2* mutant); **c** PDX #201 (HR-DNA repair gene wild-type); and **d** PDX #29 (HR-DNA repair gene wild-type). Recipient mice bearing PDX were randomized to treatment with vehicle or rucaparib, at the dose shown. PDX were harvested at a tumor volume of 600–700 mm$^3$. Cisplatin response data for PDX #19, #56, and #29 were previously published[34]. See Table 1 and Supplementary Data 2 for median TTH and *p*-values for survival comparison. Mean tumor volume (mm$^3$) ± 95% CI (hashed lines are representing individual mice) and corresponding Kaplan–Meier survival analysis. Censored events are represented by crosses on Kaplan–Meier plot; *n* = individual mice

RAD51 foci formation assay, which showed formation of RAD51 foci in response to DNA damage in PDX #169, which harbored heterozygous methylation, but not in PDX #11 or PDX #62, which both harbored homozygous methylation (Fig. 3c, d, Supplementary Fig. 7).

One of the two chemo-naive PDX with homozygous *BRCA1* promoter methylation (#62) responded to 300 mg kg$^{-1}$ rucaparib, with tumor regressions observed in two of seven mice (median TTH 71 days vs. vehicle 18 days, *p* < 0.001, log-rank test, *n* = 7, 11) (Table 1, Fig. 3e). This was notable, given that PDX #62 was resistant/refractory to cisplatin (defined as three or more mice with tumor progressing during cisplatin treatment) and was

characterized by the presence of multiple oncogene amplifications (Table 1)[34]. The other PDX with homozygous *BRCA1* methylation (#11), despite being exquisitely sensitive to platinum[34], failed to respond to rucaparib at the low dose tested (150 mg kg$^{-1}$) but was not exposed to 300 and 450 mg kg$^{-1}$ (Supplementary Data 2, Supplementary Fig. 6d). However, the patient from whom PDX #11 was derived subsequently received single-agent rucaparib with starting dose of 600 mg twice daily and had a partial response (PR) of 10 months as demonstrated by Response Evaluation Criteria In Solid Tumors version 1.1 (RECIST 1.1) (Fig. 3f, g), suggesting that in the PDX, a higher dose of rucaparib treatment in vivo may have been efficacious.

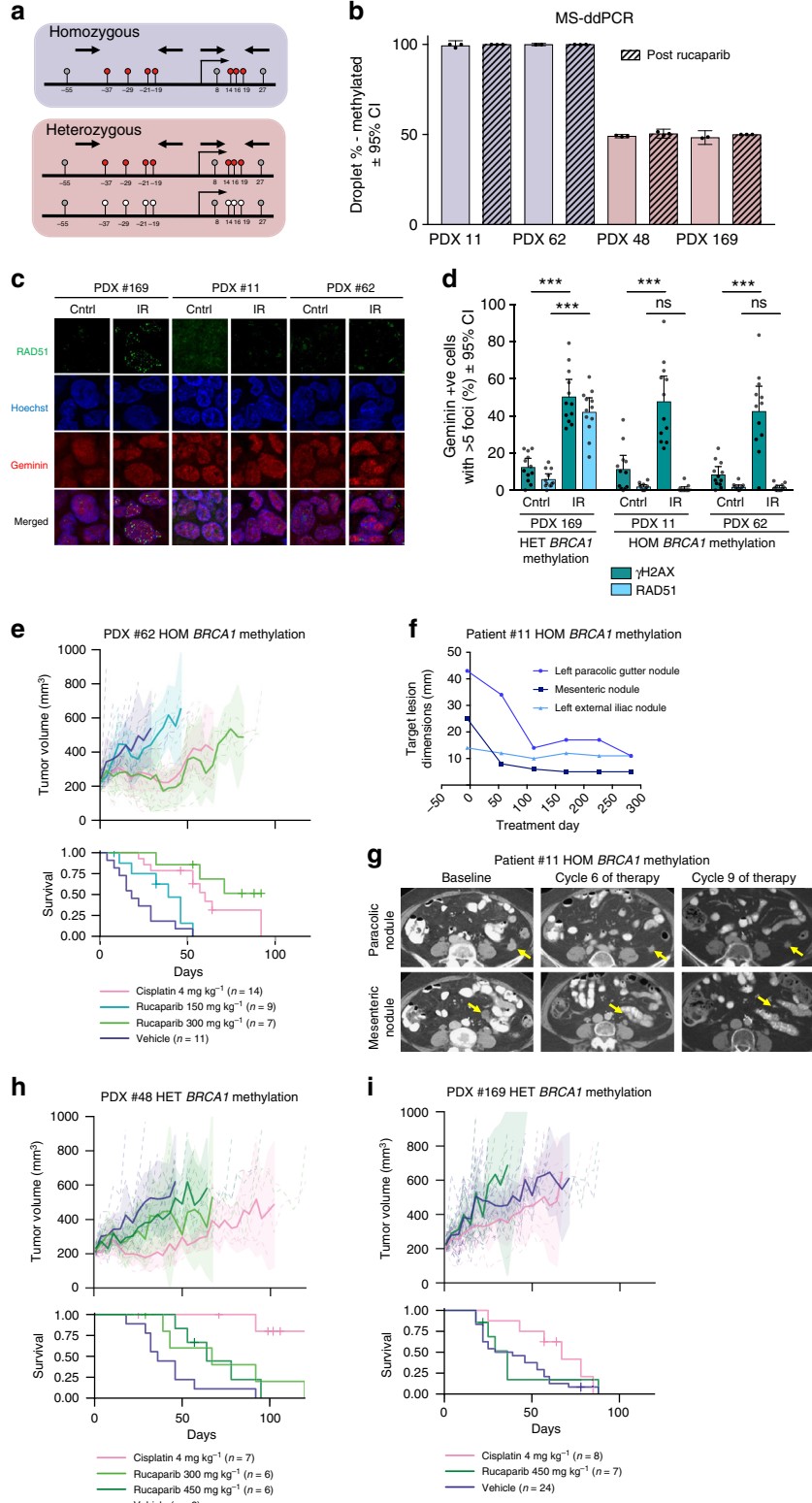

Conversely, no disease stabilization or tumor regression was observed in response to rucaparib for either PDX #48 or #169 both of which harbored heterozygous *BRCA1* promoter methylation (Table 1, Fig. 3h, i). In terms of the corresponding patient courses, case #48 progressed 4 months following third-line platinum therapy, at which point the disease was biopsied (PDX established) and she was subsequently treated with single-agent PARPi therapy, progressing after just 2 months. Case #169 had

platinum refractory disease and progressed within 1 month of first-line carboplatin/paclitaxel chemotherapy (PDX established from ascites) and did not receive a PARPi. Importantly, the heterozygous *BRCA1* methylation status in both cases reflected a change from baseline. DNA from archival patient HGSOC samples for cases #48 and #169 revealed homozygous *BRCA1* methylation in chemotherapy-naive ascites collected at diagnosis for both cases and also in the surgical debulking samples

**Fig. 3** *BRCA1* promoter methylation in HGSOC PDX and rucaparib response. **a** A diagram of two modes of *BRCA1* promoter methylation observed in four PDX #11, #62, #48, and #169. Homozygous methylation status was assigned when % of methylation was close to 100%, therefore all observed copies were methylated. Heterozygous methylation status was assigned when both, methylated and unmethylated, copies were observed. **b** *BRCA1* methylation in four HGSOC PDX (#62, #48, #169, #11) assessed by MS-ddPCR (mean ± 95% CI); $n = 2$–3 mice for each treatment and PDX model. **c** RAD51 foci formation 4 h after 10 Gy irradiation was observed in PDX #169 with heterozygous *BRCA1* methylation and not in PDX #11 and PDX #62 with homozygous *BRCA1* methylation. **d** Quantification of ex vivo γH2AX and RAD51 foci formation in geminin-positive cells 4 h after 10 Gy irradiation (mean ± 95% CI). γH2AX foci are observed at the sites of DNA damage, and RAD51 foci are observed at the sites of HR pathway repair; $n = 12$ (four fields of view from three independent experiments) for each treatment and PDX model. Untreated and irradiated cells were compared by multiple *t*-tests for γH2AX and RAD51 foci formation. ***$p < 0.001$; ns not significant. **e** Responses to cisplatin and rucaparib in vivo treatment observed in chemo-naive PDX #62 with homozygous *BRCA1* methylation. **f** RECIST 1.1 measurements of three monitored tumor lesions in patient #11, with homozygous methylation of *BRCA1*, treated with rucaparib. **g** CT scans of the two largest monitored lesions prior to and during rucaparib treatment of the patient #11. **h, i** Responses to cisplatin and rucaparib in vivo treatment observed in PDX #48 and #169 with heterozygous *BRCA1* methylation. Recipient mice bearing PDX were randomized to treatment with vehicle or rucaparib, at the dose shown. PDX were harvested at a tumor volume of 600–700 mm³ (see Table 1 and Supplementary Data 2 for median TTH and *p*-values for survival comparison). Mean tumor volume (mm³) ± 95% CI (hashed lines are representing individual mice) and corresponding Kaplan–Meier survival analysis. Censored events are represented by crosses on Kaplan–Meier plot; $n =$ individual mice. HOM homozygous, HET heterozygous

following neoadjuvant chemotherapy for both cases (Supplementary Data 4).

*BRCA1* promoter methylation was also examined in PDX samples upon cancer recurrence following in vivo treatment with either cisplatin or rucaparib. No loss of homozygous *BRCA1* methylation was observed for either PDX #11 or #62 following treatment with rucaparib or cisplatin (Fig. 3b). Similarly, no loss of heterozygous *BRCA1* methylation or gain of homozygous methylation was observed for either PDX #48 or #169 following treatment with either cisplatin or rucaparib (Fig. 3b).

**Variable rucaparib responses in *BRCA1*-methylated cell lines**. To further study whether *BRCA1* promoter methylation predisposes OC cells to rucaparib response through loss of the HR pathway activity, we generated a cell line from PDX #62 (WEHICS62) that retained homozygous *BRCA1* methylation. Reduced expression of *BRCA1* mRNA, consistent with silencing of *BRCA1*, was observed in RNA-seq and qRT-PCR analysis of the WEHICS62 cell line (two samples), the matched PDX (six samples), and the baseline patient HGSOC sample, compared to a PDX with unmethylated wild-type *BRCA1* (Supplementary Fig. 6b, Supplementary Fig. 8). In contrast, OVCAR8, a cell line generated from a patient with OC refractory to carboplatin[38,39] and previously reported to harbor *BRCA1* methylation[40], was found to have heterozygous *BRCA1* methylation when assessed by quantitative MS-ddPCR (~66% methylation with three copies of *BRCA1*, likely two methylated copies and one unmethylated copy) (Supplementary Table 2–3, Supplementary Fig. 9). Furthermore, *BRCA1* expression was detected by qRT-PCR (Supplementary Fig. 6b). This finding was in keeping with our previous report of the ability of OVCAR8 cells to form RAD51 foci and resistance of the OVCAR8 cell line to both platinum and PARPi agents in vitro, both consistent with a competent HR pathway[17].

The homozygous *BRCA1*-methylated cell line, WEHICS62, had a reduced capacity to form RAD51 foci in response to IR damage, as did an HR-deficient cell line derivative of OVCAR8 (with *RAD51C* KO), when compared to the heterozygous *BRCA1*-methylated OVCAR8 cell line, a second HR-competent OC cell line, OV90, or a normal immortalized fallopian tube epithelial cell line, FT282[41] (Fig. 4a, b). Colony formation and cell proliferation analyses revealed that WEHICS62 cells were sensitive to rucaparib, as were the HR-deficient cell line derivative of OVCAR8 (with *RAD51C* KO) and the *BRCA2*-mutant OC cell line PEO1[42]. In comparison, the parental OVCAR8 cell line with heterozygous *BRCA1* promoter methylation and the HR-

competent PEO4 OC cell line were not sensitive to rucaparib in vitro (Fig. 4c, d).

**Rucaparib response in patients with *BRCA1*-methylated HGSOC**. To investigate whether heterozygous and homozygous *BRCA1* promoter methylation correlated with PARPi response in clinical samples, we used quantitative MS-ddPCR to analyze the archival and pre-treatment (study-entry) tumor biopsies from 21 patients who were identified to have *BRCA1*-methylated HGSOC from the cohort of 204 patients treated on the ARIEL2 Part 1 single-agent rucaparib trial[32] (detail provided in methods, Table 2, Supplementary Data 5). To determine *BRCA1* methylation percentage and zygosity status in tumor cells, the raw MS-ddPCR percentage of methylated copies was adjusted for neoplastic cellularity and *BRCA1* copy number. This adjustment was required because the expected proportion of observed unmethylated copies is dependent on the ratio of unmethylated somatic copies to tumor copies at the *BRCA1* locus. A *BRCA1* copy number of 1 was assigned in cases where LOH was consistently predicted by both the Foundation Medicine T5 test and the BROCA assay and a copy number of 1 was reported by the Foundation Medicine T5 test. Cases that had a single methylated *BRCA1* locus and deletion of the second allele were classified as homozygous for *BRCA1* methylation. Low estimated neoplastic cellularity (of 20%) precluded accurate determination of *BRCA1* methylation zygosity in five out of 32 samples (archival and/or pre-treatment) tested from the 21 cases. Archival *BRCA1* methylation zygosity status could be determined with high confidence for 17 cases: ten cases were homozygous and seven cases were heterozygous (four of which had surgery at the time of diagnosis).

In order to assess the impact of *BRCA1* methylation zygosity on PARPi response, as determined at the time of treatment commencement, we focused on ARIEL2 cases for which pre-treatment biopsy samples were available for *BRCA1* methylation analysis. For 12 of the 21 *BRCA1*-methylated cases, sufficient material from pre-treatment tumor biopsies was available (Supplementary Data 5). Eight of the 12 cases (#15–21) had homozygous *BRCA1* methylation in the pre-treatment tumor biopsy, six of these (#16–21) were high confidence calls based on neoplastic cellularity of >20% (these six cases are hereafter referred to as the homozygous *BRCA1* methylation (high confidence) subgroup). Two of the 12 cases with material available for analysis (#1, 2) had homozygous and heterozygous *BRCA1* methylation, respectively, in the archival sample, and no evidence of methylation in the pre-treatment tumor biopsy

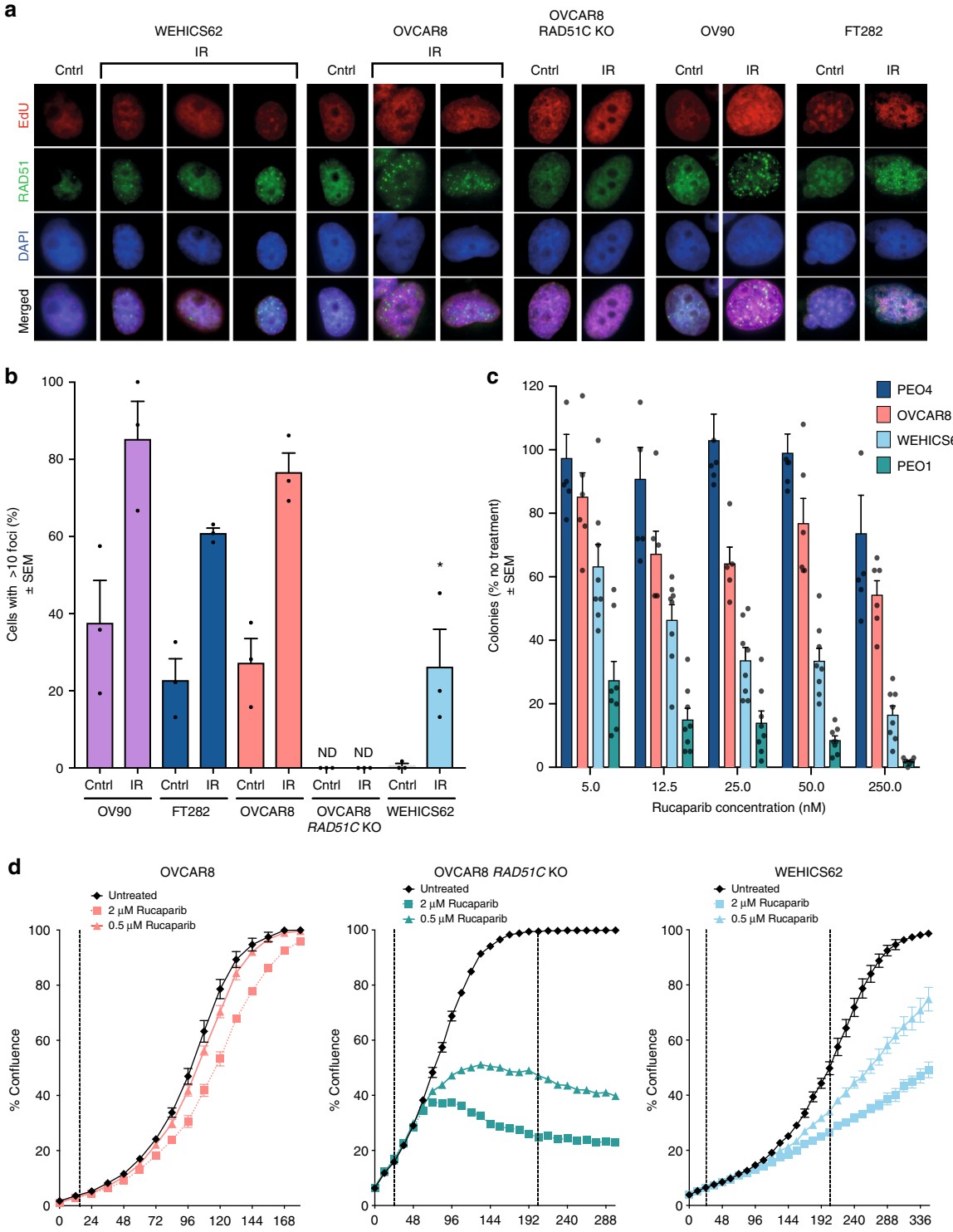

**Fig. 4** Assessment of HR deficiency and rucaparib sensitivity in *BRCA1*-methylated cell lines. **a** RAD51 foci formation assessed 6 h post exposure to 10 Gy irradiation in HR-competent OC cell line (OV90), immortalized fallopian tube cell line (FT282), OC cell line with heterozygous *BRCA1* methylation (OVCAR8), OVCAR8 derivative with *RAD51C* KO, and HGSOC cell line with homozygous *BRCA1* methylation (WEHICS62). **b** Quantification of RAD51 foci formation in EdU-positive cells for OV90, FT282, OVCAR8, OVCAR8 derivative with *RAD51C* KO, and WEHICS62. RAD51 foci formation ability was compared to the untreated controls. At least 170 EdU-positive cells were counted for each cell type and treatment (multiple fields of view from three independent experiments). Mean ± SEM. **c** Colony formation assay assessing rucaparib response at 14 days in HR-competent OC cell line (PEO4), at 10 days in OC cell line with heterozygous *BRCA1* methylation (OVCAR8) and HR-deficient OC cell line (PEO1), and at 21 days in HGSOC cell line with homozygous *BRCA1* methylation (WEHICS62); $n = 3$ independent experiments. Mean ± SEM. **d** In vitro rucaparib response assessed by cell count proliferation time course assay using IncuCyte ZOOM of OC cell lines OVCAR8, OVCAR8 derivative with *RAD51C* KO and WEHICS62. One of three similar independent experiments shown. Mean ± SEM; * denotes $p < 0.05$ for post-IR WEHICS62 % comparison with irradiated OVCAR8 and OV90 counterparts. IR irradiated, Cntrl untreated control, ND not detected

**Table 2 Degree of *BRCA1* methylation in HGSOC where a pre-treatment biopsy was available for analysis in the ARIEL2 Part 1 clinical trial**

| Patient # | Archival sample | | | | | Pre-treatment biopsy | | | | | PFS (months) | Best confirmed response |
|---|---|---|---|---|---|---|---|---|---|---|---|---|
| | BRCA1me status | Estimated BRCA1me[a] | Neoplastic cellularity | BRCA1 CN | LOH FM/ BROCA | BRCA1me status | Estimated BRCA1me[a] | Neoplastic cellularity | BRCA1 CN | LOH FM/ BROCA | | |
| 1 | HOM | 84% | 60% | 2 | Yes/yes | NO | 0% | 50% | 2 | Yes/– | 20.1 | SD |
| 2 | HET | 45% | 70.6% | 2 | Yes/– | NO | 0% | 34% | 4 | Yes/– | 1.8 | PD |
| 7 | HET[b] | 13% | 40% | 2 | Yes/– | HET | 55% | 30% | 2 | Yes/no | 14.2 | PR |
| 8 | HET | 41% | 68.9% | 2 | Yes/no | HET | 34% | 63.1% | 2 | Yes/no | 16.1 | SD[c] |
| 14 | – | – | – | – | – | HOM[d] | 77.1% | 20% | NA | –/– | 7.7 | CR |
| 15 | HOM | 75.3% | 70% | 1 | Yes/– | HOM[d] | 76.4% | 20% | 1 | Yes/– | 3.6 | SD |
| 16 | HET[b,d] | 8.4% | 20% | NA | –/– | HOM | 74.6% | 64.2% | 1 | Yes/– | 18.3 | PR |
| 17 | HET | 34.9% | 33.5% | NA | –/yes | HOM | 74.0% | 62.4% | 1 | Yes/no | 4.7 | PR |
| 18 | HOM | 98.5% | 55.8% | 2 | Yes/– | HOM | 85.6% | 64.2% | 3 | Yes/– | 17.2 | PR |
| 19 | HOM | 92.4% | 60% | 2 | Yes/yes | HOM | 101.2% | 83.6% | 2 | Yes/yes | 14.5 | SD |
| 20 | HOM | 99.3% | 52.3% | 1 | Yes/yes | HOM | 91.7% | 30% | 2 | Yes/– | 14.6 | PR |
| 21 | HOM | 86.5% | 92.7% | 1 | Yes/– | HOM | 76.4% | 64% | 1 | Yes/– | 7.2 | PR |

Neoplastic cellularity and *BRCA1* copy number were based on the computational genome-wide copy number estimates, as outlined previously[52]. Italics—low confidence calls, bold—high confidence calls
BRCA1me *BRCA1* promoter methylation, CN copy number, LOH loss of heterozygosity, FM Foundation Medicine T5 test, BROCA—BROCA assay, PFS progression-free survival, HET heterozygous, HOM homozygous, NA not available, PR partial response, PD progressive disease, SD stable disease, CR complete response
[a]If both LOH estimations (BROCA and FM) were available and concordant, we estimated BRCA1 methylation % using copy number and neoplastic cellularity, otherwise we used neoplastic cellularity
[b]Low *BRCA1* methylation %
[c]Ongoing without response
[d]Low neoplastic cellularity

(Table 2), consistent with loss of methylation. Two final cases (#7, 8) had heterozygous *BRCA1* methylation in both the archival sample and matched pre-treatment tumor biopsy.

We hypothesized that patients with homozygous *BRCA1* methylation (high confidence) at the time of enrollment into the trial ($n = 6$) would respond similarly to the *BRCA1/2* mutant subgroup with higher response rates and a longer PFS than patients with *BRCA1/2* wild-type tumors that had never been observed to have any *BRCA1* methylation (*BRCA1/2* wild-type non-*BRCA1*-methylated). The homozygous *BRCA1* methylation (high confidence) subgroup had a median PFS of 14.5 months (95% CI 4.8–18.3, $n = 6$) comparable to the *BRCA1/2* mutant subgroup (12.8 months, 95% CI 9.0–14.7, $n = 40$). Whilst not statistically significant, the PFS was longer for the high-confidence homozygous *BRCA1*-methylated group when compared to *BRCA1/2* wild-type non-*BRCA1*-methylated cases (5.5 months, 95% CI 5.0–6.2, $p = 0.062$, log-rank test, $n = 143$) (Fig. 5a).

All homozygous *BRCA1* methylation samples had high genomic LOH scores (>16%), an indirect marker of potential HR deficiency through genomic scarring[6] (Fig. 5b, Supplementary Fig. 10a). The mean LOH score for homozygous *BRCA1*-methylated cases was significantly higher than for *BRCA1/2* wild-type non-*BRCA1*-methylated cases (27.9 vs. 15.9, $p = 0.04$, independent *t*-test). There were no significant differences observed in the mean genomic LOH scores between the homozygous *BRCA1* methylation subgroup and the subgroup of other cases which had ever had any *BRCA1* methylation (Table 2, Fig. 5b, Supplementary Fig. 10a), indicating that these samples may have harbored homozygous *BRCA1* methylation in the past, leading to accumulation of genomic scarring.

The objective response rates in the high confidence homozygous *BRCA1*-methylated subgroup were significantly better compared to *BRCA1/2* wild-type non-*BRCA1*-methylated cases ($n = 143$ cases, $p = 0.0014$, Fisher Exact test, Supplementary Table 4), with five of six (83%) patients with homozygous *BRCA1* methylation detected in the pre-treatment biopsy achieving a PR and the sixth patient having a 33% reduction in target lesions that was not confirmed by subsequent CT scanning and was classified instead as stable disease (SD). A significant difference in reduction in the mean change in target lesion sizes was observed between the homozygous *BRCA1*-methylated groups and *BRCA1/2* wild-type non-*BRCA1*-methylated cases (−53.9 vs. −13.5%, $p = 0.001$, independent *t*-test, Fig. 5c, d, Supplementary Fig. 10b).

These findings indicated that cases with confirmed homozygous *BRCA1* promoter methylation were more likely to respond to rucaparib.

## Discussion

To improve our understanding of the sensitivity and resistance mechanisms to PARP inhibitors, both primary and acquired, we assessed the in vivo rucaparib response of 12 chemo-naive or post-treatment HGSOC PDX. Variable responses to rucaparib were observed in the four chemo-naive HGSOC PDX with *BRCA1/2* mutations, ranging from complete response (CR) to progressive disease. Of note, PARPi response has not been tested in the chemo-naive setting in the clinic, but our results were in keeping with the range of single-agent PARPi responses reported for patients with recurrent *BRCA1/2* mutant HGSOC[1,32,43]. We also explored mechanisms of acquired PARPi resistance by screening post-rucaparib treated PDX samples for the presence of reversion or secondary *BRCA1/2* mutations and none were observed. None of the four PDX thought to be HR proficient showed regression or disease stabilization in response to rucaparib treatment in vivo.

Recently, the ARIEL2 Part 1 trial reported that OC with *BRCA1* promoter methylation had increased levels of genomic LOH, a historical marker of HR deficiency, with some durable responses being reported[32]. To further investigate whether *BRCA1* methylation sensitized HGSOC to PARPi, we focused on the four PDX harboring this epigenetic lesion. PDX models have the advantage that the human component of PDX samples is highly enriched for neoplastic cellularity in comparison with baseline patient samples. Two of the three orthogonal *BRCA1* methylation assays used in this study were quantitative; either semi-quantitative (MS-HRM) or fully quantitative (MS-ddPCR), and both were human specific. As a result, we were able to observe two states of *BRCA1* methylation, homozygous and heterozygous, in the four *BRCA1*-methylated PDX studied. When we took into account the zygosity status of *BRCA1* methylation—which importantly has not been systematically addressed in the literature with respect to clinical outcomes and association with response to PARPi or platinum[29,33]—we observed that the zygosity of each of the four *BRCA1*-methylated PDX correlated with the zygostity status of the source tumor used to generate PDX, and did not change under the pressure of subsequent in vivo treatment of the PDX. The two chemo-naive HGSOC and corresponding PDX with homozygous *BRCA1* methylation showed

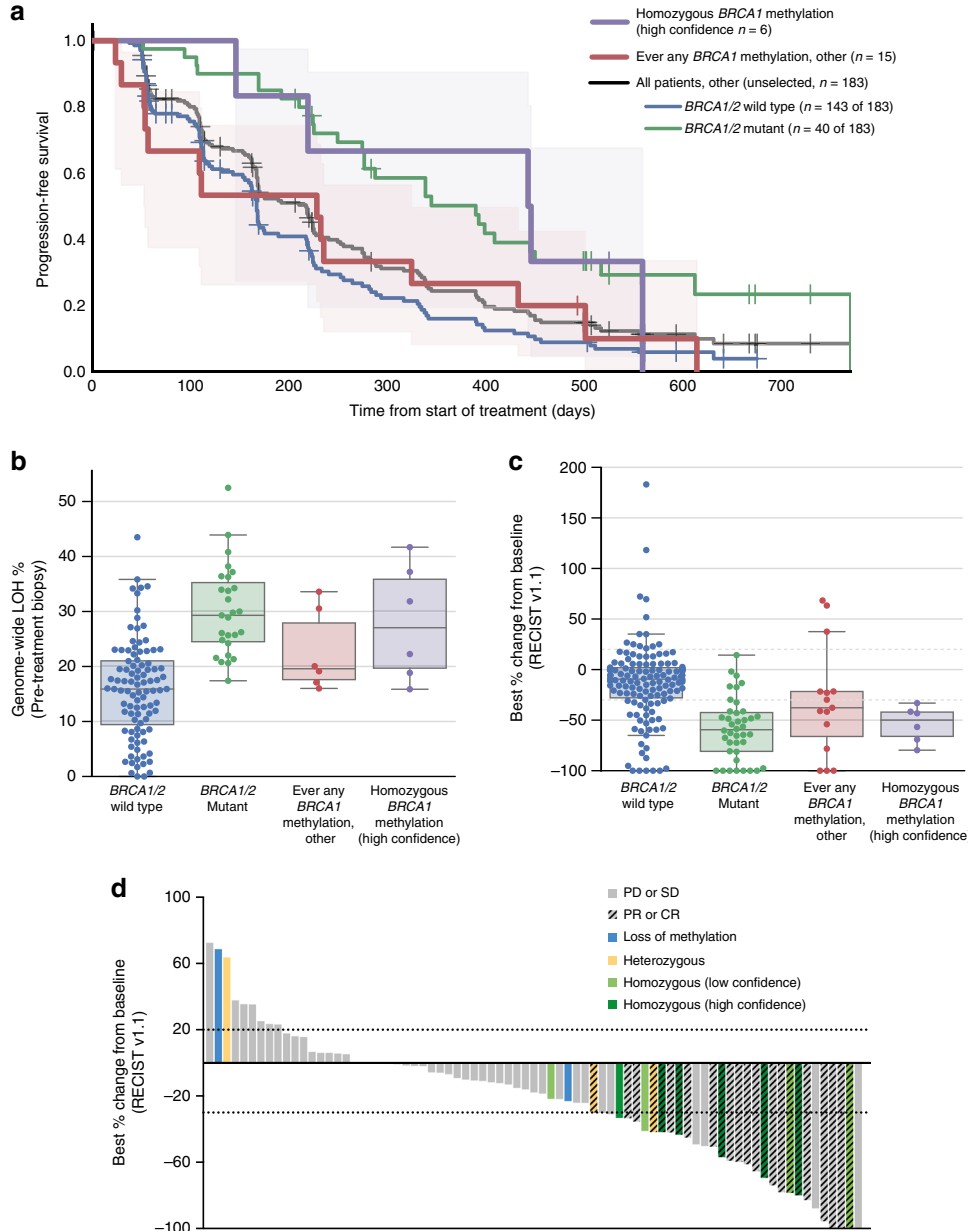

**Fig. 5** Homozygous *BRCA1* methylation and rucaparib response in the ARIEL2 Part 1 trial. **a** Kaplan–Meier progression-free survival analysis of patients with HGSOC with homozygous *BRCA1* methylation in the pre-treatment tumor biopsy, which was of high confidence based on adequate neoplastic cellularity (homozygous *BRCA1* methylation (high confidence)), compared with patients with HGSOC in which there had ever been any other evidence of *BRCA1* methylation (ever any *BRCA1* methylation), compared with all other patients in the ARIEL2 Part 1 trial without any *BRCA1* methylation (*BRCA1/2* mutant vs. *BRCA1/2* wild-type non-*BRCA1*-methylated subgroups). Shaded areas represent 95% CI for homozygous *BRCA1* methylation (high confidence) and ever any BRCA1 methylation, other groups. **b** Genome-wide LOH % assessed in the pre-treatment biopsies compared across subgroups: homozygous *BRCA1* methylation (high-confidence), (*n* = 6); ever any *BRCA1* methylation, (*n* = 6); *BRCA1/2* mutant, (*n* = 27); and *BRCA1/2* wild-type non-*BRCA1*-methylated, (*n* = 96). Boxplot—median, whiskers—95% CI, dots represent individual samples. **c** Best percentage change from baseline in sum of longest diameter of target lesions according to RECIST 1.1 compared across subgroups: homozygous *BRCA1* methylation (high confidence), (*n* = 6); ever any *BRCA1* methylation, (*n* = 15); *BRCA1/2* mutant, (*n* = 40); and *BRCA1/2* wild-type non-*BRCA1*-methylated, (*n* = 143). Boxplot—median, whiskers—95% CI, dots represent individual samples. **d** Best percentage change from baseline in sum of longest diameter of target lesions according to RECIST 1.1 in the *BRCA* wild-type LOH-high subgroup of patients by *BRCA1* methylation status. Each bar represents percentage change from baseline in sum of the longest diameter of target lesions for an individual patient according to RECIST 1.1. In some patients, although best percentage change of >30% was observed, the response was not investigator confirmed and thus classified as stable disease (SD) or progressive disease (PD). PR partial response, PD progressive disease, SD stable disease, CR complete response

low *BRCA1* expression by RNA-seq and responded to rucaparib in either the PDX or patient. Conversely, the two PDX with heterozygous *BRCA1* methylation and some expression of *BRCA1* by RNA-seq, had been generated from HGSOC treated with multiple lines of prior therapy and had failed to respond to rucaparib. In keeping with the possibility that partial loss of methylation may have occurred under treatment pressure, analysis of chemo-naive archival HGSOC samples for these two cases indicated homozygous *BRCA1* methylation. These PDX studies, coupled with earlier reports correlating complete *BRCA1* methylation and LOH at the *BRCA1* locus[21], provide evidence that, as is the case for *BRCA1* mutated carcinomas[26], silencing of all copies of *BRCA1* by promoter methylation is required to cause an HR defect of sufficient magnitude to induce PARPi-related synthetic lethality.

Consistent with these findings in PDX, we also observed that *BRCA1* promoter methylation zygosity influenced PARPi response in OC cell lines. The OVCAR8 cell line has been reported to be methylated at the *BRCA1* promoter, with reduction in *BRCA1* mRNA and protein expression[40,44], but nevertheless has been reported to be HR competent[45]. In keeping with this paradox, we found that despite having a genomic profile consistent with genomic scarring and HR deficiency[46], OVCAR8 cells formed RAD51 foci in response to DNA damage and were resistant to PARPi in vitro, consistent with an intact HR pathway. Notably, the *BRCA1* methylation status of OVCAR8 was heterozygous. In contrast, we generated a HGSOC PDX-derived cell line, WEHICS62, which retained homozygous *BRCA1* methylation, had reduced ability to form RAD51 foci and was sensitive to rucaparib in vitro. The low number of cell lines reported to date with homozygous loss of *BRCA1* or loss of function by mutation[40,47] suggests possible selection against *BRCA1*-deficient cells in two-dimensional culture. Thus, the zygosity status of *BRCA1*-methylated cell lines should be ascertained and reconfirmed regularly in studies where the HR status is critical.

In order to investigate *BRCA1* promoter methylation zygosity in clinical samples, we studied 21 high-grade OC with *BRCA1* methylation from the ARIEL2 Part 1 trial[32]. Establishing the zygosity of *BRCA1* methylation in patient samples was more challenging than in PDX, due to the variability in the proportion of normal stroma within each clinical sample, as well as variation in *BRCA1* gene copy number. We were able to identify six high-grade OC, in which we determined with high confidence that homozygous *BRCA1* methylation was present at the time of trial enrollment (in the pre-rucaparib treatment biopsy, with neoplastic cellularity >20%). Consistent with our hypothesis that homozygous methylation was required for sensitivity to rucaparib, we observed a strong association between homozygous *BRCA1* methylation and rucaparib response, compared to *BRCA1/2* wild-type non-*BRCA1*-methylated cases. It would thus be important for patient selection/stratification to use a highly quantitative method for methylation assessment, as well as accurate estimation of neoplastic cellularity and *BRCA1* copy number to determine the zygosity of *BRCA1* methylation, even at diagnosis, when *BRCA1*-methylated cases might already have heterozygous rather than homozygous methylation. Indeed, our data strongly suggest that methylation zygosity should be assessed in contemporaneous tumor samples before any firm conclusions are drawn regarding the impact of *BRCA1* methylation on therapeutic efficacy[29,33].

Genomic scarring assays, such as percentage genomic LOH score, are indicative of defective HR DNA repair and are likely to identify HR-defective cases with homozygous *BRCA1* methylation. However, there will also be false-positive cases, where HR has been restored (e.g., through loss of methylation) yet the genomic scarring remains, reflecting a history of prior HR

deficiency rather than the current HR status. In keeping with this, LOH status (percentage genomic LOH score) was not different in the *BRCA1* homozygous methylated vs. "ever-methylated" cases in the ARIEL2 Part 1 trial. Furthermore, we identified cases in our clinical studies that supported the hypothesis that loss of methylation of the *BRCA1* promoter could occur under platinum treatment pressure. In our PDX studies, we observed altered zygosity of *BRCA1* methylation, from homozygosity in the chemo-naive archival clinical sample to heterozygosity in the previously-treated HGSOC patient source sample used to generate the PDX. Thus, heterozygous *BRCA1*-methylated cases may represent loss of methylation from an earlier homozygous *BRCA1* methylation and HR-deficient state, contributing to PARPi resistance.

A survival advantage has been demonstrated for *BRCA1/2* mutated HGSOC treated with platinum-based therapies[7], despite reversion of *BRCA1/2* mutation occurring under treatment pressure[14–16]. However, a number of challenges may exist in demonstrating a survival advantage for *BRCA1* promoter-methylated HGSOC. The first is that our data suggest that cases need to be classified according to *BRCA1* methylation zygosity. The second is that homozygous *BRCA1* methylation loss may occur readily under chemotherapy pressure, as our analysis of the ARIEL2 data suggests, necessitating a pre-treatment biopsy. Confirmatory studies of *BRCA1* methylation zygosity in larger clinical cohorts will also be required; however, large PARPi trials in OC have focused on the maintenance setting, where tumor tissue immediately prior to PARPi has not been collected. The ARIEL2 Part 1 study is the only study to date which has routinely collected pre-treatment biopsies and which is of sufficient size to permit this analysis (204 patients enrolled with the expected case rate of *BRCA1* methylation of ~10%, yielding 21 *BRCA1*-methylated cases, of which 12 had homozygous methylation in the archival biopsy, eight of which had homozygous methylation in the pre-treatment biopsy (six of which were of adequate tumor purity)). Given the difficulty in accessing additional similar or larger cohorts, our data support early scheduling of PARPi treatment, for example in the first-line maintenance setting, or as combination PARPi therapy upon first relapse, to minimize the population of malignant cells in which loss of methylation and consequently resistance to treatment can occur.

In summary, the study of PDX models, in which quantitative assessment of tumor *BRCA1* promoter methylation is not obscured by stromal signal or complicated by variable copy number, enabled the observation that homozygous methylation and complete silencing of *BRCA1* induces HR deficiency and PARPi sensitivity. By using a highly quantitative method and adjusting for neoplastic cellularity and *BRCA1* copy number, we identified patients with homozygous *BRCA1* methylation in the ARIEL2 Part 1 PARPi study and observed improved clinical outcomes. This study is the first to clarify the critical role of *BRCA1* methylation zygosity in the response of carcinomas to PARPi and has potentially important clinical implications. Further development and refinement of methods to accurately and efficiently classify *BRCA1* homozygous methylation status, and a suitably powered prospective clinical trial are required to validate our findings that homozygous *BRCA1* methylation predicts PARPi sensitivity. Further, *BRCA1* methylation loss in carcinomas exposed to chemotherapy underscores the importance of real-time pre-treatment biopsies to assess methylation as a predictor of response to treatment in women with recurrent HGSOC. As with secondary mutations in *BRCA1/2*, loss of methylation of *BRCA1* under treatment pressure disables a therapeutic mechanism of response, suggesting that earlier introduction of PARPi therapy in the disease trajectory may

prevent such resistance mechanisms emerging under treatment pressure.

## Methods

**Study approval**. All experiments involving animals were performed according to the animal ethics guidelines and were approved by the Walter and Eliza Hall Institute of Medical Research Animal Ethics Committee. PDX were generated from OC, with patients enrolled in the Australian Ovarian Cancer Study. Informed consent was obtained from all patients, and all experiments were performed according to the human ethics guidelines. Additional ethics approval was obtained from the Human Research Ethics Committees at the Royal Women's Hospital and the Walter and Eliza Hall Institute.

**Patient samples**. Surgical, biopsy or ascites HGSOC samples used for PDX generation were collected from chemotherapy-naive patients who underwent surgery or patients treated with multiple lines of prior therapy. Clinical follow-up of patient outcome was obtained via the database at the Royal Women's Hospital. Archival tumor and pre-treatment biopsy samples from 23 patients used for re-testing of *BRCA1* promoter methylation were collected as part of the ARIEL2 Part 1 trial (NCT01891344)[32]. All clinical information used for interrogating rucaparib response in ARIEL2 Part 1 participating patients was collected as part of the ARIEL2 Part 1 trial[32]. Patient response was assessed according to RECIST 1.1.

**Generation and treatment of PDX**. PDX were generated as published previously by transplanting fresh fragments subcutaneously or via the intra-ovarian bursal approach into NOD/SCID/IL2Rγnull recipient mice (T1, passage 1)[34], with the exception of PDX #169, which was generated from tumor ascites. Briefly, tumor cells were isolated from ascites after centrifugation and red blood cell lysis. The tumor cells were resuspended in diluted Matrigel Matrix (Corning) and were subcutaneously injected.

Recipient mice bearing T2–T9 (passage 2 to passage 9) tumors were randomly assigned to treatment with rucaparib, cisplatin or vehicle when tumor volume reached 180–300 mm³. In vivo cisplatin treatments were performed as previously described[34]. The regimen for rucaparib treatment was oral gavage once daily (Monday–Friday) for 3 weeks at 150, 300, or 450 mg kg⁻¹. Tumors were measured twice per week and recorded in StudyLog software (StudyLog Systems). Tumors were harvested once tumor volume reached 600–700 mm³ or when mice reached ethical endpoint. Nadir, time to progression (TTP or PD), TTH, and treatment responses are as defined previously[34]. Tumor volume and survival graphs were produced with SurvivalVolume v1.2[48]. Median TTH was calculated by including censored events for PDX where mice were harvested when the tumor volume was >500 mm³ but <600 mm³ (for rucaparib response for PDX #62, 4 out of 6 mice). CR was achieved if the average tumor volume for the treatment group reduced to <50 mm³ for two or more consecutive measurements. PR was achieved if the average tumor volume reduced to between 50 and 140 mm³ (>30% reduction from nadir, assigned as 200 mm³) for two or more consecutive measurements. SD was achieved if TTP for the treatment group was at least twice as long as TTP for the corresponding vehicle group.

**Cell lines and culture**. The human OC cell line OVCAR8 was obtained from the NCI. Early passages of the parental OVCAR8 and RAD51C KO 2–130 were cryopreserved, and were last authenticated by STR profiling in April 2017. Subsequent revivals were used within 6 months. OC cell line WEHICS62 was generated from PDX #62, by digesting cells with human tumor dissociation kit (Mitenyl Biotec) with gentleMACS dissociator, and then enriching for viable Epcam (347197 1:30; BD) positive cells using flow cytometry. Early-passage were viably stored; subsequent thaws were used within 6 months. The STR profile for WEHICS62 was generated in April 2017: Amelogenin—X; CSF1PO—allele 12; D13S317—allele 9; D16S539—allele 13; D21S11—alleles 29, 30; D5S818—allele 7; D7S820—allele 9; TH01—allele 7; TPOX—allele 8; and vWA—alleles 17, 18. The PEO4 and PEO1 cell lines were obtained from F. Couch (Mayo Clinic) in 2013 and viably stored until 2016. Subsequent thaws were used within 6 months; were last authenticated by STR profiling in April 2017. All cell lines were routinely tested and shown to be negative for *Mycoplasma*. Cell lines were cultured in RPMI-1640 (Corning) supplemented with 10% FBS (Peak Serum) and 1% penicillin and streptomycin (Corning) or in DMEM/F12, GlutaMAX with 5 μg ml⁻¹ insulin, 50 ng ml⁻¹ EGF, and 1 μg ml⁻¹ hydrocortisone in a 5% CO₂ atmosphere at 37 °C. FT282 cells were grown in DMEM:Ham's F12 (50:50) without HEPES in the presence of Ultroser G serum substitute.

**Compounds**. Rucaparib camsylate salt was manufactured by Lonza. Cisplatin was obtained from Pfizer.

**RAD51 foci formation assay**. For RAD51 foci assay in cell lines, cells were first treated with 10 mM EdU, then shortly after irradiated with 10 Gy. Cells were fixed 6 h post irradiation with 10% paraformaldehyde, permeabilized with 0.3% TritonX-100, blocked with blocking buffer (5% goat serum, 0.3% TritonX-100 in PBS), and incubated with rabbit anti-human RAD51 (ab63801 1:100; Abcam), followed by

incubation with goat anti-rabbit 488 secondary antibody (1:600; Invitrogen). Cells were incubated for 30 min at room temperature in Click-IT reaction (100 mM Tris pH 8.5, 10 nM Alexa Fluor 647-azide (Cat# A10277, Thermo Fisher Scientific), 1 mM CuSO₄, and 100 mM ascorbic acid), then washed with PBS. Nuclei were counterstained with DAPI in Vectashield mounting media (Vector Labs). Images were acquired on an Olympus BX-61 microscope equipped with a Spot RT camera (model 25.4), using the Spot Advanced software. EdU positive cells with more than 10 RAD51 foci/nucleus were manually scored. At least 170 cells from three independent experiments were counted.

For ex vivo RAD51 foci assay, tumor tissue was first harvested, then placed in cell culturing medium and shortly after irradiated with 10 Gy or left untreated. Tissue fragments were fixed 4 h post irradiation for 2 h with 4% paraformaldehyde, then incubated in 10, 20, and 30% sucrose, embedded in Tissue-Tek® O.C.T. (optimal cutting temperature) compound (Sakura Finetek), and 4 μm sections were cut. Following antigen retrieval with pH 6 citrate buffer (Dako) in a pressure cooker, sections were permeabilized for 20 min with 0.2% Triton-X, washed in DPBS, and blocked for 30 min with blocking buffer (1% bovine serum albumin, 2% fetal bovine serum in DPBS). Sections were incubated overnight at 4 °C with rabbit anti-human RAD51 (ab133534 1:100; Abcam) or rabbit anti-human γH2AX (20E3 1:200; Cell Signaling), washed with DPBS, then incubated for 1 h at room temperature (RT) with anti-rabbit 488 secondary antibody (1:800; Invitrogen), washed with DPBS, then incubated for 1 h at RT with mouse anti-human Geminin (ab104306 1:100; Abcam), washed with DPBS, then incubated for 1 h at RT with anti-mouse 546 secondary antibody (1:800, Invitrogen) and Hoechst (1 drop ml⁻¹), then washed with DPBS and mounted with Fluoromount-G® (SouthernBiotech). All antibody dilutions were done with blocking buffer. Sections were imaged using a LSM 780 inverse laser scanning microscope (Zeiss) and captured with an LSM T-PMT detector (Zeiss) using z-stacks. Z-stacks were flattened using Z projection function with maximum intensity in Fiji software. At least 230 cells from four fields of view and three independent experiments were counted. Cells with ≥5 RAD51 or γH2AX foci/geminin-positive nucleus were scored using CellProfiler (version 2.2.0, Broad Institute).

**Cell proliferation assay**. Cell count proliferation assay was performed using IncuCyte ZOOM system according to the manufacturer's protocol. Briefly, cells were seeded (OVCAR8 at 500 cells; OVCAR8-RAD51C KO and WEHISC62 at 2000 cells) in 96-well plates and incubated overnight before adding treatments. Cells were treated for up to 14 days with rucaparib at 0.5 or 2 μM or medium. Medium with rucaparib or without rucaparib was replenished at 7 days.

**Colony formation assay**. Colony formation assay was performed on OC cell lines PEO4, PEO1, OVCAR8, and WEHICS62. Briefly, cells were seeded at 100 cells in 6-well plates and incubated overnight before adding treatments. Cells were treated with rucaparib at 5, 12.5, 25, 50, and 250 nM or equivalent amount of DMSO (no treatment control). The experiment was terminated when colonies formed in cultures without treatment (PEO4 cells fixed at 14 days; OVCAR8 and PEO1 cells fixed at 10 days; WEHICS62 cells fixed at 21 days). Colonies were fixed with 0.5% Crystal Violet and 20% methanol for 20 min. Colonies were counted blindly by three individuals, and then the average count was taken for each replicate.

**Immunohistochemistry**. Staining was performed using an automated platform with a DAKO Omnis (Agilent Pathology Solutions) on all first-generation PDX samples (T1) to confirm the retention of HGSOC characteristics when compared to the clinical pathology report, or in-house staining, at the time of sample collection. The following antibodies were used: p53 (M700101 1:100; Dako), Ki67 (M7240 1:50; Dako), Cytokeratin (M3515 1:200; Dako), PAX8 (10336–1-AP 1:20000; Proteintech), and WT1 (ab15249 1:800; Abcam). CD45 (M0701 1:500; Dako) was used to exclude donor-derived transplantable hematologic malignancy. Scoring was performed for each PDX by one investigator on one tumor section each from at least three independent mice bearing that PDX and from the relevant baseline patient tumor. Usually ten high-powered fields (for some only five were available) were surveilled and the staining estimated as follows: 3+ almost all tumor cells were strongly positive; 2+ >25% of tumor cells were strongly positive or nearly all tumor cells were moderately positive; 1+ <25% of tumor cells were moderately to strongly positive, or nearly all cells were weakly positive; 0 occasional positive cells only.

**Genomic analyses and qRT-PCR**. RNA-seq was performed on 12 baseline patient HGSOC samples used to generate PDX, and on two PDX samples (#169 and #201) to verify expression levels observed in the matched HGSOC with suboptimal sample quality due to either low neoplastic cellularity or poor RNA quality. Libraries were prepared using TruSeq RNA Library Prep Kit v2, and the sequencing was performed on the Ilumina HiSeq 2500 platform to read length of 50 bp (Australian Genome Research Facility). Reads were mapped to Human GRCh38 (GCA_000001405.15) with dbSNP150 and Ensembl 90 annotation using HISAT2[49], and annotated against dbSNP150 and Ensembl 90. Counts were done using HTSeq[50], and TMM normalization was performed[51]. Expression plots were produced using Matplotlib. Baseline patient HGSOC sample #80 had to be

excluded from final analysis as it failed alignment QC due to high proportion of multi-mapping reads.

For qRT-PCR *BRCA1* assay, RNA was converted to cDNA using Superscript III Reverse Transcriptase (Invitrogen), and qPCR was performed using SYBR Green PCR Master Mix (Applied Biosystems) following manufacturer's instructions. Primer sequences are listed in Supplementary Table 5. Ct values for each sample were normalized to the average Ct values of four different housekeeping genes (*HPRT1*, *ACTB*, *SDHA*, and *GAPDH*), and resulting values were used to calculate fold-change of *BRCA1* expression for each sample.

Baseline patient HGSOC samples used to generate PDX or PDX samples were sequenced using Foundation Medicine's NGS-based T5a assay[52]. Analyzed data were plotted using OncoPrint. HR-DNA repair gene mutations were assessed by sequencing of PDX samples using the NGS-based BROCA assay: PDX #11, #13, #27, #29, #56, #62 were analyzed using BROCA v4 assay and were previously published[34]; and all others were analyzed by BROCA v6 (Supplementary Data 6).

Copy number analysis of *BRCA1* was performed using MLPA-seq assay as previous described[53]. Amplicon sequencing of *TP53* gene was performed on the patient HGSOC and PDX samples to estimate the neoplastic cellularity proportion. Amplicon libraries were prepared and sequenced as previously described[17], with primers listed in the Supplementary Table 5. Samples from the four *BRCA1/2* mutant HGSOC were screened for reversion mutations by NGS and Sanger sequencing. Sanger sequencing was performed to amplify target regions in PDX #13, #19, #54, and #56. Long-range PCR was used to amplify target regions in PDX #19 and #56 samples, with primers specified in Supplementary Table 5. Briefly, 100–120 ng of DNA was amplified with TaKaRa LA Taq polymerase (Takara Bio Inc) or Phusion Polymerase (Thermo Fisher Scientific) using the following cycling conditions: initial denaturation at 94 °C for 1 min or 98 °C for 30 s (respectively), followed by 30 cycles of 94 °C or 98 °C for 15 s, 60–62.8 °C for 30 s or 64–58 °C (−0.2 °C sec⁻¹) for 20 s (respectively), and 68 °C for 15 min or 72 °C for 3 min (respectively), followed by final extension at 72 °C for 10 min. Long-range PCR products were cleaned using Agencourt AMPure XP (Beckman Coulter) beads at 1:0.4 ratio following the manufacturer's protocol then processed using the Nextera XT DNA Library Preparation Kit (Illumina) according to manufacturer's protocol. The libraries were sequenced using a 300-cycle MiSeq Nano Reagent Kit v2 (Illumina).

**Promoter methylation analysis**. Promoter methylation of *BRCA1* PDX samples was determined by methylation-sensitive PCR as previously described[21]. *BRCA1* methylation was confirmed by MS-HRM, as previously described[24]. Quantification of *BRCA1* methylation levels in PDX sample series, cell lines, and ARIEL2 Part 1 patient sample series was assessed by a quantitative MS-ddPCR methodology. DNA was bisulfite converted using the EZ DNA Methylation-Lightning kit (Zymo Research). Primers were designed for a 72 bp amplicon in the *BRCA1* UTR. MGB probes hybridizing to the fully methylated (VIC labeled) and the fully unmethylated sequences (FAM labeled) were used. Droplet digital PCR was performed on the Bio-Rad QX-200 system.

**Western blotting**. Nuclear lysates were prepared from snap frozen tumor fragments. Western blotting was carried out using NE-PER Nuclear and Cytoplasmic Extraction Reagents (Thermo Scientific) as previously described[54] and proteins were detected using the following antibodies: BRCA1 (OP92-MS110 1:500, Calbiochem) and Tubulin (2148 1:2000, Cell Signaling).

**Analysis of ARIEL2 Part 1 clinical trial cases**. Of 204 patients included in the ARIEL2 Part 1 clinical trial and treated with single-agent rucaparib, 23 cases had evidence of *BRCA1* promoter methylation according to prior MSP analysis[32]. For these 23 cases, we assessed *BRCA1* methylation in a second DNA extraction from the same tumor sample by MS-ddPCR. Two of the 23 cases were excluded from the *BRCA1*-methylated subgroup because assessment did not confirm *BRCA1* methylation. Of the 21 samples with *BRCA1* methylation by MS-ddPCR, pre-treatment biopsies were available for 12 cases, eight of which had homozygous *BRCA1* methylation, and six of these were of high confidence based on adequate neoplastic cellularity (high-confidence homozygous *BRCA1* methylation, $n = 6$). The other 15 cases were included in the "ever any *BRCA1* methylation" subgroup as they had other evidence of *BRCA1* methylation, which was not high-confidence homozygous methylation in the pre-treatment biopsy. These cases were compared to all other HGSOC from the ARIEL2 Part 1 trial without any evidence of *BRCA1* methylation (204 cases, minus 21 cases = 183 cases), subdivided by *BRCA1/2* mutant ($n = 40$ cases) and *BRCA1/2* wild-type (*BRCA1/2* wild-type non-*BRCA1*-methylated $n = 143$ cases) status.

**Statistics**. Statistical analysis was performed to compare the high-confidence homozygous *BRCA1*-methylated subgroup of patients (homozygous *BRCA1* methylation (high confidence), $n = 6$ cases) with the subgroup of cases with any other evidence of *BRCA1* methylation (ever any *BRCA1* methylation, $n = 15$ cases), or with the *BRCA1/2* mutant subgroup ($n = 40$) or the *BRCA1/2* wild-type non-*BRCA1*-methylated subgroup ($n = 143$). Statistical analysis was performed in

Python 3.6.1 using the pandas v0.20.2, lifelines v0.11.1, seaborn v0.8.1, matplotlib v2.0.2, and scipy v0.19.1 packages. Comparisons were made between high-confidence homozygous *BRCA1*-methylated subgroup of patients and the *BRCA1/2* wild-type non-methylated subgroup. No statistical comparison was made between the high-confidence homozygous *BRCA1*-methylated subgroup of patients and the ever any *BRCA1* methylation subgroup, as the latter group contains low confidence homozygous and ambiguous cases. Fisher Exact test was used to compare the investigator-confirmed best response, independent *t*-test was used to compare the minimum percentage change of target lesion, and Kaplan–Meier analysis and log rank test were used to compare PFS.

## Data availability

Access to data including RNA-seq can be requested through Walter and Eliza Hall Institute of Medical Research Data Access Committee by contacting dataaccess@wehi. edu.au. The data are not publicly available due to them containing information that could compromise research participant privacy and consent.

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

## Acknowledgements

We thank Dr. M. Herold for the Cas9/CRISPR vectors used to generate PEO4 *RAD51C* KO, and T. Phuong, S. Stoev, R. Hancock, K. Barber, and C. Mitchell for technical assistance. We also thank Dr. Michael Christie for reviewing the archival *BRCA1*-methylated HGSOC samples. The AOCS also acknowledges the contribution of the study nurses, research assistants, and all clinical and scientific collaborators to the study. The complete AOCS Study Group can be found at www.aocstudy.org. We would like to thank all of the women who participated in these research programs. This work was supported by fellowships and grants from the National Health and Medical Research Council (NHMRC Australia; Fellowship to KJH (#494836), Project grant 1062702 (CLS)); the Stafford Fox Medical Research Foundation (CLS); Cancer Council Victoria (Sir Edward Dunlop Fellowship in Cancer Research to CLS); the Victorian Cancer Agency (Clinical Fellowships to CLS CRF10–20, CRF16014); Monash University (PhD Scholarship to MT); CRC for Cancer Therapeutics (PhD top-up scholarship to MT, GH); Research Training Program Scholarship (PhD Scholarship to KN, GH); Bev Gray Ovarian Cancer Scholarship (PhD top-up scholarship to KN, VH); NHMRC Scholarship (1076048 to AH) the National Breast Cancer Foundation (PS-15-048 to AD) and the National Institute of Health (2P50CA083636) (to EMS) and the Wendy Feuer Ovarian Cancer Research Fund (to EMS). The Olivia Newton-John Cancer Research Institute acknowledges the support of the Victorian Government Operational and Infrastructure Support Program. This work was made possible through the Australian Cancer Research Foundation, the Victorian State Government Operational Infrastructure Support and Australian Government NHMRC IRIISS. We thank Clovis Oncology for funding Foundation Medicine analyses and providing rucaparib for in vivo experiments. Funding was provided by a Stand Up To Cancer—Ovarian Cancer Research Fund Alliance—National Ovarian Cancer Coalition Dream Team Translational Research Grant (grant number SU2C-AACR-DT16-15 to EMS and SHK). Stand Up to Cancer is a programme of the Entertainment Industry Foundation; research grants are administered by the American Association for Cancer Research, a scientific partner of Stand Up To Cancer. The Australian Ovarian Cancer Study Group was supported by the U.S. Army Medical Research and Materiel Command under DAMD17-01-1-0729, The Cancer Council Victoria, Queensland Cancer Fund, The Cancer Council New South Wales, The Cancer Council South Australia, The Cancer Council Tasmania and The Cancer Foundation of Western Australia (Multi-State Applications 191, 211 and 182), and the National Health and Medical Research Council of Australia (NHMRC; ID400413 and ID400281). The Australian Ovarian Cancer Study gratefully acknowledges additional support from S. Boldeman, the Agar family, Ovarian Cancer Action (UK), Ovarian Cancer Australia and the Peter MacCallum Foundation.

## Author contributions

C.L.S., M.J.W., A.D., and E.M.S. designed the study, developed the methodology, analyzed the data, wrote the manuscript, and supervised the study. O.K. designed the study, developed the methodology, performed the experiments, analyzed the data, wrote the manuscript, and supervised the study. M.T. developed the methodology, performed the experiments, analyzed the data, and reviewed the manuscript. S.H.K. assisted in study design and wrote the manuscript. K.N., E.L., M.I.H., G.H., E.S., G.V.Z., and J.J.K. developed the methodolgy, performed the experiments, analyzed the data, and reviewed the manuscript. N.T. and K.A. acquired the data, provided administrative support, and reviewed the manuscript. A.deF., K.K.L., T.C.H., and I.A.M. acquired the data, supervised the study, and reviewed the manuscript. R.B.P., K.J.H., and N.J. supervised the study and reviewed the manuscript. A.H., R.H., E.B., V.H., O.M., S.A., D.D.B., and AOCS acquired the data and reviewed the manuscript. K.N. and E.L. contributed equally to this work.

## Additional information

**Competing interests:** K.K.L. and T.C.H. are employees of Clovis Oncology, Inc. and hold stock in Clovis Oncology. A.deF. has received research grant support from AstraZeneca. The remaining authors declare no competing interests.

Olga Kondrashova[1,2], Monique Topp[1,3], Ksenija Nesic[1,2], Elizabeth Lieschke[1], Gwo-Yaw Ho[1,2,4,5], Maria I. Harrell[6], Giada V. Zapparoli[7,8], Alison Hadley[1,2], Robert Holian[1,9], Emma Boehm[1,9], Valerie Heong[1,2,4], Elaine Sanij[5,10], Richard B. Pearson[5,11,12,13], John J. Krais[14], Neil Johnson[14], Orla McNally[4], Sumitra Ananda[4], Kathryn Alsop[5], Karla J. Hutt[3], Scott H. Kaufmann[15], Kevin K. Lin[16], Thomas C. Harding[16], Nadia Traficante[5,10], Australian Ovarian Cancer Study (AOCS), Anna deFazio[17], Iain A. McNeish[18], David D. Bowtell[5,10], Elizabeth M. Swisher[6], Alexander Dobrovic[7,8,10], Matthew J. Wakefield[1,19] & Clare L. Scott[1,3,5]

[1]The Walter and Eliza Hall Institute of Medical Research, Parkville, VIC 3052, Australia. [2]Department of Medical Biology, University of Melbourne, Parkville, VIC 3010, Australia. [3]Department of Medicine and Health Sciences, Monash University, Clayton, VIC 3168, Australia. [4]Royal Women's Hospital, Parkville, VIC 3052, Australia. [5]Research Division, Peter MacCallum Cancer Centre, Grattan Street, Parkville, VIC 3010, Australia. [6]Department of Obstetrics and Gynecology, University of Washington, Seattle, WA 98195, USA. [7]Olivia Newton-John Cancer Research Institute, Heidelberg, VIC 3084, Australia. [8]School of Cancer Medicine, La Trobe University, Bundoora, VIC 3086, Australia. [9]School of Medicine, University of Melbourne, Parkville, VIC 3052, Australia. [10]Department of Clinical Pathology, University of Melbourne, Melbourne, VIC 3010, Australia. [11]Sir Peter MacCallum Department of Oncology, The University of Melbourne, Melbourne, VIC 3010, Australia. [12]Department of Biochemistry and Molecular Biology, University of Melbourne, Parkville, VIC 3010, Australia. [13]Department of Biochemistry and Molecular Biology, Monash University, Clayton, VIC 3168, Australia. [14]Fox Chase Cancer Centre, Philadelphia, PA 19111, USA. [15]Departments of Oncology and Molecular Pharmacology, Mayo Clinic, Rochester, MN 55905, USA. [16]Clovis Oncology, Boulder, CO 80301, USA. [17]Centre for Cancer Research, The Westmead Institute for Medical Research, Sydney Medical School, The University of Sydney and Department of Gynaecological Oncology, Westmead Hospital, Sydney, NSW 2145, Australia. [18]Division of Cancer, Department of Surgery and Cancer, Imperial College London, Kensington, London SW7 2AZ, United Kingdom. [19]Melbourne Bioinformatics, University of Melbourne, Melbourne, VIC 3010, Australia. These authors contributed equally: Olga Kondrashova, Monique Topp. These authors jointly supervised this work: Alexander Dobrovic, Matthew J. Wakefield, Clare L. Scott. A full list of consortium members appears below.

## Australian Ovarian Cancer Study (AOCS)

G. Chenevix-Trench[20], A. Green[20], P. Webb[20], D. Gertig[21], S. Fereday[22], S. Moore[20], J. Hung[23], K. Harrap[20], T. Sadkowsky[20], N. Pandeya[20], M. Malt[20], A. Mellon[24], R. Robertson[24], T. Vanden Bergh[25], M. Jones[25], P. Mackenzie[25], J. Maidens[26], K. Nattress[27], Y.E. Chiew[23], A. Stenlake[23], H. Sullivan[23], B. Alexander[20], P. Ashover[20], S. Brown[20], T. Corrish[20], L. Green[20], L. Jackman[20], K. Ferguson[20], K. Martin[20], A. Martyn[20], B. Ranieri[20], J. White[28], V. Jayde[29], P. Mamers[30], L. Bowes[22], L. Galletta[22], D. Giles[22], J. Hendley[22], T. Schmidt[31], H. Shirley[31], C. Ball[32], C. Young[32], S. Viduka[31], H. Tran[31], S. Bilic[31], L. Glavinas[31], J. Brooks[33], R. Stuart-Harris[34], F. Kirsten[35], J. Rutovitz[36], P. Clingan[37], A. Glasgow[37], A. Proietto[24], S. Braye[24], G. Otton[24], J. Shannon[38], T. Bonaventura[39], J. Stewart[39], S. Begbie[40], M. Friedlander[41], D. Bell[26], S. Baron-Hay[26], A. Ferrier[26], G. Gard[26], D. Nevell[26], N. Pavlakis[26], S. Valmadre[26], B. Young[26], C. Camaris[25], R. Crouch[25], L. Edwards[25], N. Hacker[25], D. Marsden[25], G. Robertson[25], P. Beale[27], J. Beith[27], J. Carter[27], C. Dalrymple[27], R. Houghton[27], P. Russell[27],

M. Links[42], J. Grygiel[43], J. Hill[44], A. Brand[45], K. Byth[45], R. Jaworski[45], P. Harnett[45], R. Sharma[46], G. Wain[45], B. Ward[47], D. Papadimos[47], A. Crandon[48], M. Cummings[48], K. Horwood[48], A. Obermair[48], L. Perrin[48], D. Wyld[48], J. Nicklin[48,49], M. Davy[28], M.K. Oehler[28], C. Hall[28], T. Dodd[28], T. Healy[50], K. Pittman[31], D. Henderson[51], J. Miller[52], J. Pierdes[33], P. Blomfield[29], D. Challis[29], R. McIntosh[29], A. Parker[29], B. Brown[53], R. Rome[53], D. Allen[54], P. Grant[54], S. Hyde[54], R. Laurie[54], M. Robbie[54], D. Healy[30], T. Jobling[30], T. Manolitsas[30], J. McNealage[30], P. Rogers[30], B. Susil[30], E. Sumithran[30], I. Simpson[30], K. Phillips[22], D. Rischin[22], S. Fox[22], D. Johnson[22], S. Lade[22], M. Loughrey[22], N. O'Callaghan[22], W. Murray[22], P. Waring[55], V. Billson[4], J. Pyman[4], D. Neesham[4], M. Quinn[4], C. Underhill[56], R. Bell[57], L.F. Ng[58], R. Blum[59], V. Ganju[60], I. Hammond[32], Y. Leung[32], A. McCartney[32], M. Buck[61], I. Haviv[62], D. Purdie[20], D. Whiteman[20] & N. Zeps[31]

[20]QIMR Berghofer Medical Research Institute, Brisbane, QLD 4006, Australia. [21]Melbourne School of Population and Global Health, University of Melbourne, Parkville, VIC 3052, Australia. [22]Peter MacCallum Cancer Centre, East Melbourne, VIC 3002, Australia. [23]Centre for Cancer Research, The Westmead Millennium Institute for Medical Research, The University of Sydney and Departments of Gynaecological Oncology, Westmead Hospital, Sydney, NSW 2145, Australia. [24]John Hunter Hospital, Lookout Road, New Lambton, NSW 2305, Australia. [25]Royal Hospital for Women, Barker Street, Randwick, NSW 2031, Australia. [26]Royal North Shore Hospital, Reserve Road, St Leonards, NSW 2065, Australia. [27]Royal Prince Alfred Hospital, Missenden Road, Camperdown, NSW 2050, Australia. [28]Royal Adelaide Hospital, North Terrace, Adelaide, SA 5000, Australia. [29]Royal Hobart Hospital, 48 Liverpool St, Hobart, TAS 7000, Australia. [30]Monash Medical Centre, 246 Clayton Rd, Clayton, VIC 3168, Australia. [31]Western Australian Research Tissue Network (WARTN), St John of God Pathology, 23 Walters Drive, Osborne Park, WA 6017, Australia. [32]Women and Infant's Research Foundation, King Edward Memorial Hospital, 374 Bagot Road, Subiaco, WA 6008, Australia. [33]St John of God Hospital, 12 Salvado Rd, Subiaco, WA 6008, Australia. [34]Canberra Hospital, Yamba Drive, Garran, ACT 2605, Australia. [35]Bankstown Cancer Centre, Bankstown Hospital, 70 Eldridge Road, Bankstown, NSW 2200, Australia. [36]Northern Haematology & Oncology Group, Integrated Cancer Centre, 185 Fox Valley Road, Wahroonga, NSW 2076, Australia. [37]Illawarra Shoalhaven Local Health District, Wollongong Hospital, Level 4 Lawson House, Wollongong, NSW 2500, Australia. [38]Nepean Hospital, Derby Street, Kingswood, NSW 2747, Australia. [39]Newcastle Mater Misericordiae Hospital, Edith Street, Waratah, NSW 2298, Australia. [40]Port Macquarie Base Hospital, Wrights Road, Port Macquarie, NSW 2444, Australia. [41]Prince of Wales Clinical School, University of Sydney, Sydney, NSW 2031, Australia. [42]St George Hospital, Gray Street, Kogarah, NSW 2217, Australia. [43]St Vincent's Hospital, 390 Victoria Street, Darlinghurst, NSW 2010, Australia. [44]Wagga Wagga Base Hospital, Docker St, Wagga Wagga, NSW 2650, Australia. [45]Crown Princess Mary Cancer Centre, Westmead Hospital, Westmead, Sydney, NSW 2145, Australia. [46]Department of Pathology, Westmead Clinical School, Westmead Hospital, The University of Sydney, Sydney, NSW 2006, Australia. [47]Mater Misericordiae Hospital, Raymond Terrace, South Brisbane, QLD 4101, Australia. [48]The Royal Brisbane and Women's Hospital, Butterfield Street, Herston, QLD 4006, Australia. [49]Wesley Hospital, 451 Coronation Drive, Auchenflower, QLD 4066, Australia. [50]Burnside Hospital, 120 Kensington Road, Toorak Gardens, SA 5065, Australia. [51]Flinders Medical Centre, Flinders Drive, Bedford Park, South Australia 5042, Australia. [52]Queen Elizabeth Hospital, 28 Woodville Road, Woodville South, SA 5011, Australia. [53]Freemasons Hospital, 20 Victoria Parade, East Melbourne, VIC 3002, Australia. [54]Mercy Hospital for Women, 163 Studley Road, Heidelberg, VIC 3084, Australia. [55]Department of Pathology, University of Melbourne, Parkville, VIC 3052, Australia. [56]Border Medical Oncology, Nordsvan Dr and Pearce Dr, Wodonga, VIC 3690, Australia. [57]Andrew Love Cancer Centre, 70 Swanston Street, Geelong, VIC 3220, Australia. [58]Ballarat Base Hospital, Drummond Street North, Ballarat, VIC 3350, Australia. [59]Bendigo Health Care Group, 62 Lucan Street, Bendigo, VIC 3550, Australia. [60]Peninsula Health, 2 Hastings Road, Frankston, VIC 3199, Australia. [61]Mount Hospital, 150 Mounts Bay Road, Perth, WA 6000, Australia. [62]Faculty of Medicine, Bar-Ilan University, 8 Henrietta Szold St, Safed 1311502, Israel.

