## [Peer Review File · Nature Communications]

Reviewers' comments:

Reviewer #1 (Remarks to the Author):

The paper describes that BRCA1 methylation can predict response to PARP-inhibitors, providing homozygous or hemizygous methylation is observed. Partial methylation, with a concomitant partial reduction of expression, was not sufficient to produce rucaparib sensitivity. The point of the paper is clear and worthwhile, but the details still need to be more convincing to support to the definitive statement of the abstract.

The problems are in the details of the observations. First of all, there are data from BRCA1/2 mutant tumors, which we know are sensitive to PARPi (with occasional exceptions that we do not fully understand). The fully wild-type cells are generally resistant, therefore the four samples grown as PDXs with homozygous or heterozygous inactivation of BRCA1 by methylation are the key samples with respect to novel observations. Figure 3 is therefore the key figure – we need the data from PDX sample #11 to be shown in Fig 3d, otherwise the key observation is based entirely on sample #62 versus #48 and #169.

Similarly, it would be better if we had evidence of RAD51 function correlating with the completeness of methylation. Cell line data are shown and the evidence is based on knockdown of RAD51C in OVCAR8 to show that there is HR function remaining in the single allele methylation. One of the cell lines has full methylation (WEHICS62) but the response to rucaparib is significantly less than the RAD51C depleted line (Fig 4d). RAD51 foci can be assessed in PDXs, so this would help support the conclusions. The clinical data from the ARIEL2 trial is somewhat supportive of the idea that full methylation is more likely to be associated with response (Fig 5d) but the number of events driving this observation are limited.

In summary, the novel observation of the study would be to show that BRCA1 methylation status predicts for HR function and response to PARP-inhibitors. Some of the data are background context (BRCA-mutant vs wild-type) but the critical data to support the methylation status are patchy rather than complete. More robust data focusing on methylation status only is required.

Reviewer #2 (Remarks to the Author):

This manuscript examines the role that BRCA1 methylation may play in predicting response to the PARP inhibitor rucaparib in high grade serous ovarian cancer, the most common histotype. It is well written and addresses the uncertainty with the effects of BRCA1 methylation noted in the literature. Use of a novel quantitative assay for BRCA1 methylation allows determination of methylation status of both alleles of BRCA1, an important advance.

By incorporation of BRCA1 gene expression, mutational status of BRCA2 and BRCA1, genomic instability as well as methylation, the report clearly clarifies that homozygous BRCA1 methylation status does indeed indicate defective HR, and further, that if one allele is missing, BRCA1 methylation is sufficient to abrogate HR. However, if one allele is functional, even if the other is methylated, HR is intact.

The authors also note that PFS is similar between homozygous BRCA1 methylated and BRCA1/2 mutant HGSO. Notably, homozygous BRCA1 methylation may be lost under chemotherapy pressure. As most clinical trials do not collect tumor tissue prior to chemotherapy, they took advantage of the ARIEL2 Part 1 study which did collect tissue. Some ten percent of the sample showed homozygous BRCA1 methylation.

This is a well reported study, showing that assessment of BRCA1 methylation (and other HR genes) as well as mutational analysis prior to therapy may lead to better management of patients with ovarian cancer.

Reviewer #3 (Remarks to the Author):

General comments:

1. On page 26, in the "Statistics" section, for the first 8 lines, the authors mentioned that "statistical analysis was performed ...". But it is not clear what statistical tests were done for each result (p-value) presented in this paper, even though later "Fisher Exact test, t-test, and log-rank test" were mentioned in this section. For example, on page 13 and 14, the authors showed 4 p-values, but it is not clear what statistical tests were conducted to obtain those p-values. It would be very helpful for the readers if the authors can add the test beside the p-value (e.g., p-value = 0.001 based on the Fisher's exact test).

2. The author reported and conducted analysis for n=6 homozygous BRCA1 methylation samples, but there are n=40 for mutant, and n=143 for other type, does this mean it is rare to observe/get homozygous BRCA1 methylation samples? If so, how many samples would you recommend to do confirmatory studies in a larger clinical cohort (as stated on page 18 of the original manuscript)?

Reviewer #1 (Remarks to the Author):

The paper describes that BRCA1 methylation can predict response to PARP-inhibitors, providing homozygous or hemizygous methylation is observed. Partial methylation, with a concomitant partial reduction of expression, was not sufficient to produce rucaparib sensitivity. The point of the paper is clear and worthwhile, but the details still need to be more convincing to support to the definitive statement of the abstract.

We thank the Reviewer for noting that the point of our paper, that partial methylation was not sufficient to produce rucaparib sensitivity, is clear and worthwhile.

The problems are in the details of the observations. First of all, there are data from BRCA1/2 mutant tumors, which we know are sensitive to PARPi (with occasional exceptions that we do not fully understand). The fully wild-type cells are generally resistant, therefore the four samples grown as PDXs with homozygous or heterozygous inactivation of BRCA1 by methylation are the key samples with respect to novel observations. Figure 3 is therefore the key figure – we need the data from PDX sample #11 to be shown in Fig 3d, otherwise the key observation is based entirely on sample #62 versus #48 and #169.

Unfortunately, PDX #11 is not available for treatment at the higher dose of 300 mg/kg, hence we are unable to include informative PDX data in Fig 3d. Instead, we now include patient response data, as Patient #11 was treated on a PARP inhibitor clinical trial. We now present Patient #11 response to rucaparib, with new Figure 3f-g, CT images and percent change in target lesion size by RECIST v 1.1. These data clearly show that HGSOc #11 responded to treatment with rucaparib, achieving objective partial response (65% reduction at nadir) which was maintained for 10 months (PD occurred when a new lesion was observed at 10 months following dose reductions for toxicity). This supports our hypothesis that case #11 with homozygous methylation of *BRCA1* responded to treatment with a PARP inhibitor.

Similarly, it would be better if we had evidence of RAD51 function correlating with the completeness of methylation. Cell line data are shown and the evidence is based on knockdown of RAD51C in OVCAR8 to show that there is HR function remaining in the single allele methylation. One of the cell lines has full methylation (WEHICS62) but the response to rucaparib is significantly less than the RAD51C depleted line (Fig 4d). RAD51 foci can be assessed in PDXs, so this would help support the conclusions.

We agree with the Reviewer about the importance of providing evidence of RAD51 function in PDX and now provide this for three *BRCA1* methylated PDX models. We observed RAD51 foci formation in PDX #169 with heterozygous methylation and not in PDX #11 and #62 with homozygous methylation, further supporting our hypothesis that homozygous *BRCA1* methylation causes HR deficiency, while heterozygous *BRCA1* methylation results in HR proficiency.

The clinical data from the ARIEL2 trial is somewhat supportive of the idea that full methylation is more likely to be associated with response (Fig 5d) but the number of events driving this observation are limited.

Unfortunately, there are currently no other PARPi clinical trials which routinely collected pre-treatment biopsies, which are of sufficient size to permit similar analysis of *BRCA1* methylated cases (see size calculation below).

In summary, the novel observation of the study would be to show that BRCA1 methylation status predicts for HR function and response to PARP-inhibitors. Some of the data are background context (BRCA-mutant vs wild-type) but the critical data to support the methylation status are patchy rather than complete. More robust data focusing on methylation status only is required.

In response to the Reviewer's request for more robust data, we have provided six lines of new evidence.

In addition to confirming the patient response for case #11 above and providing the evidence of RAD51 function in three PDXs requested by Reviewer 1, we have also performed additional analyses:

- We now show that the case #169 with heterozygous *BRCA1* methylation retained homozygous *BRCA1* methylation until after three cycles of neoadjuvant chemotherapy treatment and only reverted to heterozygous methylation at some point after receiving three cycles of adjuvant chemotherapy and one subsequent cycle of chemotherapy post relapse (prior to collection of ascites). We did this by more accurately estimating the tumor purity in the surgical debulk patient tumor samples, by assessing the *TP53* mutation frequency in the same DNA aliquots as the ones used for *BRCA1* methylation analysis. Prior to this we had assessed tumor purity by H&E but needed to set up a new *TP53* sequencing assay in order to determine tumor purity with more accuracy. Thus, both of the heterozygously methylated cases were homozygously methylated until after first relapse, relevant for consideration of sequencing of PARPi therapy in the clinic.
- To strengthen our finding that the OVCAR8 cell line has 3 copies of *BRCA1*, two methylated and one unmethylated, we performed *BRCA1* MS-ddPCR on single cell colonies, which showed consistent ~66% methylation. We also performed copy number analysis on whole cell line extract, which confirmed that OVCAR8 has three copies of *BRCA1*.
- We performed *BRCA1* qRT-PCR on OVCAR8 to show *BRCA1* expression in comparison to reduced expression in WEHICS-62 with homozygous *BRCA1* methylation.
- We also assessed *BRCA1* protein expression levels by WB in the four PDX models with *BRCA1* methylation, which supported our hypothesis that *BRCA1* is silenced in homozygously methylated cases, whereas it is expressed in heterozygously methylated cases. This assessment was performed blinded as to sample identity by Neil Johnson's group at the Fox Chase Cancer Center who have specific expertise in *BRCA1* western blotting.

Reviewer #2 (Remarks to the Author):

This manuscript examines the role that BRCA1 methylation may play in predicting response to the PARP inhibitor rucaparib in high grade serous ovarian cancer, the most common histotype. It is well written and addresses the uncertainty with the effects of BRCA1 methylation noted in the literature. Use of a novel quantitative assay for

BRCA1 methylation allows determination of methylation status of both alleles of BRCA1, an important advance.

We thank the Reviewer for noting that our manuscript addresses uncertainty in the literature and that the use of a novel quantitative assay for BRCA1 methylation is an important advance.

By incorporation of BRCA1 gene expression, mutational status of BRCA2 and BRCA1, genomic instability as well as methylation, the report clearly clarifies that homozygous BRCA1 methylation status does indeed indicate defective HR, and further, that if one allele is missing, BRCA1 methylation is sufficient to abrogate HR. However, if one allele is functional, even if the other is methylated, HR is intact. The authors also note that PFS is similar between homozygous BRCA1 methylated and BRCA1/2 mutant HGSOV. Notably, homozygous BRCA1 methylation may be lost under chemotherapy pressure. As most clinical trials do not collect tumor tissue prior to chemotherapy, they took advantage of the ARIEL2 Part 1 study which did collect tissue. Some ten percent of the sample showed homozygous BRCA1 methylation. This is a well reported study, showing that assessment of BRCA1 methylation (and other HR genes) as well as mutational analysis prior to therapy may lead to better management of patients with ovarian cancer.

Reviewer 2 required no additional experiments.

Reviewer #3 (Remarks to the Author):

General comments:

1. On page 26, in the "Statistics" section, for the first 8 lines, the authors mentioned that "statistical analysis was performed ...". But it is not clear what statistical tests were done for each result (p-value) presented in this paper, even though later "Fisher Exact test, t-test, and log-rank test" were mentioned in this section. For example, on page 13 and 14, the authors showed 4 p-values, but it is not clear what statistical tests were conducted to obtain those p-values. It would very helpful for the readers if the authors can add the test beside the p-value (e.g., p-value =0.001 based on the Fisher's exact test).

We thank the Reviewer for this suggestion and have now added the statistical tests used next to each p-value listed in the main text.

2. The author reported and conducted analysis for n=6 homozygous BRCA1 methylation samples, but there are n=40 for mutant, and n=143 for other type, does this mean it is rare to observe/get homozygous BRCA1 methylation samples? If so, how many samples would you recommend to do confirmatory studies in a larger clinical cohort (as stated on page 18 of the original manuscript)?

To estimate how many patients would need to be screened for larger confirmatory studies, a power calculation was performed by Matthew Maurer, statistician at the Mayo Clinic.

The assumptions were: a 1 sided test with alpha of 0.05 and 90% power, 3 years to accrue and 2 years of follow-up, median survival in group 1 is 430 days, median survival in group 2 is 225 days, 28.6% of patients are in group 1, and survival follows an exponential distribution with no dropouts.

The estimations were: a total sample size of approximately 900 patients would need to be screened, with 91 patients with BRCA1 methylation required to have 90% power to detect a difference.

It would be extremely challenging to complete a trial of this size and duration at this time, as the field has moved on to ask different types of questions, involving combination therapies for the most part. It is extremely unlikely that patients would be willing to take part in a clinical trial offering them only single agent therapy when PARP inhibitors, including rucaparib, are now FDA approved for maintenance therapy regardless of BRCA status. Instead, it is important that our findings be available for consideration by the field so that we can continue to develop rational strategies based on accurate science to promote patient-/tumour-specific algorithms within the clinic so that patients can access the types of treatments most relevant for their needs.

REVIEWERS' COMMENTS:

Reviewer #1 (Remarks to the Author):

The revised paper, which describes that BRCA1 methylation can predict response to PARP-inhibitors, providing that homozygous or hemizygous methylation is observed, is now improved with better functional evidence that the methylation status and repair pathway function do correlate well. Last time, we requested the data from PDX sample #11 to be shown in Fig 3, otherwise the key observation would be based entirely on sample #62 versus #48 and #169. This has now been provided in the form of the actual patient response rather than the PDX. The authors have also provided evidence of RAD51 focus formation (function) correlating with the completeness of methylation.

The novel observation of this study is that BRCA1 methylation status predicts for HR function and response to PARP-inhibitors. The data are now more robust in relation to methylation status, which makes the paper significantly improved.

Reviewer #2 (Remarks to the Author):

I believe that the authors have adequately addressed raised by the reviewers

Reviewer #3 (Remarks to the Author):

Thanks for addressing my questions.